# Dissemination Routes of Carbapenem and Pan-Aminoglycoside Resistance Mechanisms in Hospital and Urban Wastewater Canalizations of Ghana

Jose F. Delgado-Blas,[a,b] Cecilia Valenzuela Agüí,[a*] Elena Marin Rodriguez,[a] Carlos Serna,[a,b] Natalia Montero,[a,b] Courage Kosi Setsoafia Saba,[c] Bruno Gonzalez-Zorn[a,b]

[a]Animal Health Department, Faculty of Veterinary Medicine, Complutense University of Madrid, Madrid, Spain
[b]VISAVET Health Surveillance Centre, Complutense University of Madrid, Madrid, Spain
[c]Department of Biotechnology, Faculty of Agriculture, University for Development Studies, Tamale, Ghana

**ABSTRACT** Wastewater has a major role in antimicrobial resistance (AMR) dynamics and public health. The impact on AMR of wastewater flux at the community-hospital interface in low- and middle-income countries (LMICs) is poorly understood. Therefore, the present study analyzed the epidemiological scenario of resistance genes, mobile genetic elements (MGEs), and bacterial populations in wastewater around the Tamale metropolitan area (Ghana). Wastewater samples were collected from the drainage and canalizations before and after three hospitals and one urban waste treatment plant (UWTP). From all carbapenem/pan-aminoglycoside-resistant bacteria, 36 isolates were selected to determine bacterial species and phenotypical resistance profiles. Nanopore sequencing was used to screen resistance genes and plasmids, whereas, sequence types, resistome and plasmidome contents, pan-genome structures, and resistance gene variants were analyzed with Illumina sequencing. The combination of these sequencing data allowed for the resolution of the resistance gene-carrying platforms. Hospitals and the UWTP collected genetic and bacterial elements from community wastewater and amplified successful resistance gene-bacterium associations, which reached the community canalizations. Uncommon carbapenemase/$\beta$-lactamase gene variants, like $bla_{DIM-1}$, and novel variants, including $bla_{VIM-71}$, $bla_{CARB-53}$, and $bla_{CMY-172}$, were identified and seem to spread via clonal expansion of environmental *Pseudomonas* spp. However, $bla_{NDM-1}$, $bla_{CTX-M-15}$, and *armA* genes, among others, were associated with MGEs that allowed for their dissemination between environmental and clinical bacterial hosts. In conclusion, untreated hospital wastewater in Ghana is a hot spot for the emergence and spread of genes and gene-plasmid-bacterium associations that accelerate AMR, including to last-resort antibiotics. Urgent actions must be taken in wastewater management in LMICs in order to delay AMR expansion.

**IMPORTANCE** Antimicrobial resistance (AMR) is one the major threats to public health today, especially resistance to last-resort compounds for the treatment of critical infections, such as carbapenems and aminoglycosides. Innumerable works have focused on the clinical ambit of AMR, but studies addressing the impact of wastewater cycles on the emergence and dissemination of resistant bacteria are still limited. The lack of knowledge is even greater when referring to low- and middle-income countries, where there is an absence of accurate sanitary systems. Furthermore, the combination of short- and long-read sequencing has surpassed former technical limitations, allowing the complete characterization of resistance genes, mobile genetic platforms, plasmids, and bacteria. The present study deciphered the multiple elements and routes involved in AMR dynamics in wastewater canalizations and, therefore, in the local population of Tamale, providing the basis to adopt accurate control measures to preserve and promote public health.

Address correspondence to Bruno Gonzalez-Zorn, bgzorn@ucm.es.

*Present address: Cecilia Valenzuela Agüí, Department of Biosystems Science and Engineering, Eidgenössische Technische Hochschule Zürich, Zürich, Switzerland.

The authors declare no conflict of interest.

**KEYWORDS** antibiotic resistance, environmental microbiology, genomic analyses, plasmid-mediated resistance, public health, wastewater treatment

Wastewater is formed by heterogeneous material from diverse sources and provides a favorable medium in which numerous biological processes take place. Furthermore, the resulting compounds and biological associations could reach the environment and, eventually, human and animal populations due to the flow of wastewater. Thus, the correct treatment of wastewater is one of the cornerstones for public health preservation (1). However, low- and middle-income countries (LMICs) largely lack the infrastructures and systems to carry out the accurate processing of wastewater, not only from the community, but also from hospitals and health care settings, which entails a greater health risk (2). Additionally, the absence of suitable canalizations favors contact with untreated wastewater and its circulation among the population, and, consequently, the potential dissemination of bacteria (3).

Many previous studies have exposed the role of wastewater, especially from health care settings, as a source and vehicle of multidrug-resistant (MDR) bacteria (4, 5). Nonetheless, the antimicrobial resistance (AMR) scenario in wastewater from LMICs is still poorly understood (6). Moreover, the deficiency in antibiotic prescription and usage control measures facilitates the selection and fixation of resistance determinants in the bacterial population, even to compounds considered last-resort antibiotics and whose administration is restricted in other regions (7). This is the case for carbapenems and aminoglycosides, which are included in the "High Priority" category of the "Critically Important Antimicrobials for Human Medicine" due to their efficacy against MDR *Enterobacteriaceae* infections with limited therapeutic options (8–10). However, the understanding about the prevalence and dissemination of resistance mechanisms to these antibiotics in the environment is limited, especially in LMICs, and studies that address this are required (6, 11).

The "Critical Priority" section of the "Global Priority List of Antibiotic-Resistant Bacteria to Guide Research, Discovery, and Development of New Antibiotics" is constituted by clinically relevant bacterial species resistant to carbapenems: *Acinetobacter baumannii*, *Pseudomonas aeruginosa*, and *Enterobacteriaceae* (12). Carbapenem-resistant (CR) bacteria can escape carbapenem activity by diverse mechanisms, including overexpression of specific $\beta$-lactamases, porin modifications, or elimination by efflux pumps. However, the most prevalent mechanisms are the carbapenemases. These enzymes are able to hydrolyze carbapenem molecules and are intrinsic resistance determinants in specific bacterial species. Some carbapenemases can be encoded in the bacterial chromosome, while others are integrated in mobile genetic elements (MGEs), which allow horizontal dissemination between different bacterial clones, species, and genera (13). Aminoglycosides are often prescribed together with $\beta$-lactam compounds, including carbapenems, due to their synergistic activity. The most prevalent aminoglycoside resistance mechanisms worldwide are the aminoglycoside-modifying enzymes, a large group of different enzymes that degrade very specific compounds of this group of antibiotics and confer moderate levels of resistance. However, 16S rRNA methyltransferases (16S RMTases), a limited group of enzymes that add a methyl group on a particular residue of the bacterial ribosome, thus blocking the attachment of aminoglycosides to their intracellular target, are able to confer high levels of resistance to most clinically relevant aminoglycosides, even to the last-resort aminoglycoside plazomicin (14). The association of some 16S RMTase genes with diverse MGEs has increased their potential for spread over the last decade, even between bacteria from human, animal, and environmental sources. Furthermore, these MGEs frequently cointegrate 16S RMTase genes with other resistance determinants, especially $\beta$-lactamase genes, due to the broad use of aminoglycoside–$\beta$-lactam combinations in human medicine (15).

Therefore, the present study was conducted in order to identify the presence of bacteria highly resistant to carbapenems or/and aminoglycosides in the wastewater of urban and hospital canalizations in the region of Tamale, Ghana. Concurrently, we aimed to analyze the genetic mechanisms responsible for these antimicrobial resistances and the MGEs involved in their dissemination in order to decipher the epidemiological scenario and flux of these bacterial elements across the hospital-community interface.

## RESULTS

**Hospital and UWTP environments acted as amplifiers and gatherers of MDR bacteria.** Samples were obtained from each of the four sampling locations—three hospitals (Tamale West Hospital [TWH], Tamale Central Hospital [TCH], and Tamale Teaching Hospital [TTH]) and one urban waste treatment plant (UWTP) located in the metropolitan area of Tamale. Wastewater samples were obtained from three different points: the canalization before the hospital (the first treatment pond in the case of the UWTP), the drainage of the hospital onto the urban canalization (the last treatment pond of the UWTP), and the canalization after the hospital (the point where treated water is discharged to the environment after the UWTP) (Fig. 1A; see Table S1 in the supplemental material). From each point, duplicate samples were collected, for a total of 24 wastewater samples. Total bacterial counts showed a remarkable increase in samples from hospital drainages and the UWTP last pond, compared with the samples from canalizations before and after the hospitals and the UWTP, which displayed lower sums of total bacteria (Fig. 1B). Likewise, carbapenem- and/or pan-aminoglycoside-resistant bacteria also exhibited a notable expansion in most of the hospital/UWTP environments, in accordance with the trend of total bacterial growth. However, the counts of resistant bacteria in wastewater from canalizations after hospitals remained higher than those observed from canalizations before the hospital (Fig. 1B): from 0.00069 to 0.013%, from 0.0 to 0.00036% and from 0.0262 to 0.133% carbapenem/pan-aminoglycoside-resistant bacteria in TWH, TCH, and TTH, respectively. Therefore, the amplification of resistant bacteria in hospital environments led to the release of these bacteria to later urban canalizations. In the case of the TTH, both the total bacteria and the resistant bacteria found in all samples presented considerably high and stable counts, even in wastewater from canalizations before the hospital (Fig. 1B). This increasing trend toward resistant bacteria across the canalizations of the region can be explained by the wastewater flux and course, as will be discussed later.

**The absence of hospital-community boundaries allowed the spread of MDR bacteria across the region.** From the screening with carbapenem/aminoglycoside-supplemented media, up to three carbapenem/pan-aminoglycoside-resistant colonies, when available, were selected, obtaining a total of 36 isolates. Among these, the most prevalent bacterial species identified via matrix-assisted laser desorption ionization–time of flight mass spectrometry (MALDI-TOF MS) were *Pseudomonas putida* (11 isolates [30.5%]) and *Citrobacter werkmanii* (11 isolates [30.5%]). The remaining 14 isolates were distributed between 10 different bacterial species, including other members of the *Pseudomonas* and *Citrobacter* genera and typical environmental bacteria, such as *Providencia rettgeri*. PCRs of predominant carbapenemase and 16S RMTase genes revealed $bla_{NDM-1}$ and *armA* as the mechanisms responsible for carbapenem and pan-aminoglycoside resistance, respectively, in *Enterobacteriaceae* species—mainly in *C. werkmanii* isolates. In *P. rettgeri*, the responsible genes were $bla_{NDM-1}$ and *rmtC*. However, no genes were detected in carbapenem-resistant *Pseudomonas* spp., nor in pan-aminoglycoside-resistant *Comamonas* and *Delftia* species isolates. Nanopore sequencing was carried out with 25 selected isolates belonging to different species, in order to perform a whole-genome screening for total resistance gene content and further structure resolution. This screening uncovered the association of carbapenem resistance in *Pseudomonas* spp. with a carbapenemase gene belonging to the $bla_{VIM}$ family and the presence of resistance gene families to multiple antibiotics in all isolates, including those previously detected by PCR. All 36 carbapenem/pan-aminoglycoside-resistant isolates were sequenced by Illumina technology to study bacterial clonality and sequence type, resistance gene variants, and plasmid types. Based on AMR gene content, the resistance profiles showed a high correlation with bacterial species, independent of the sample origin, except for *P. aeruginosa*, which exhibited a variable resistance gene content (Fig. 2) (see Microreact at https://microreact.org/project/2sBABj7YhfhoLZi8trXvSv).

*P. putida* was detected in all hospital sampling locations and wastewater canalizations and displayed a multidrug-resistant profile, remaining susceptible only to colistin from all tested antibiotics. MLST revealed the presence of two novel *P. putida* sequence types, denominated as ST125 and ST126. All *P. putida* isolates were resistant to carbapenems, mainly due to the presence of variants of the carbapenemase gene $bla_{VIM}$. Regarding resistance to

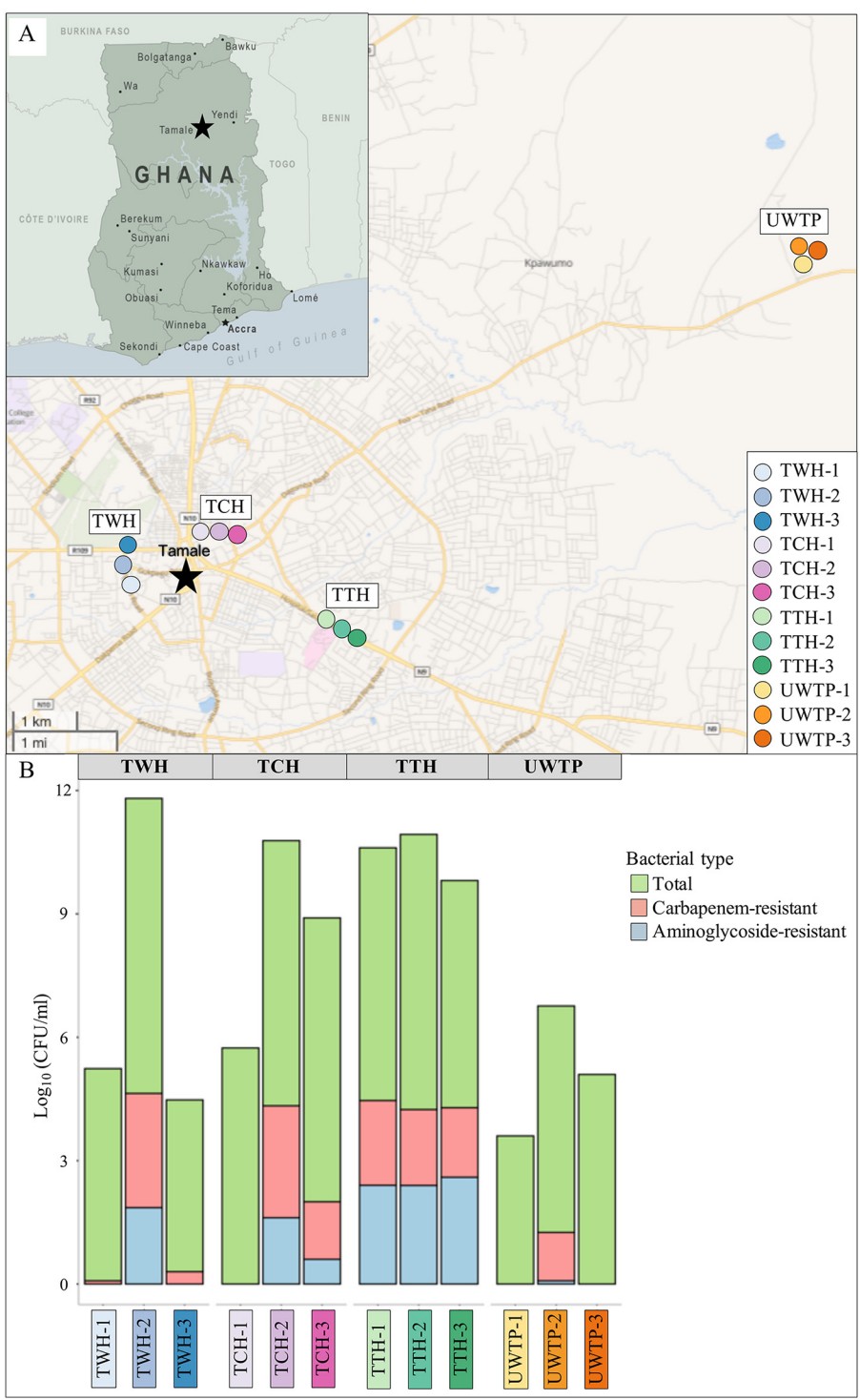

**FIG 1** Geographical locations and bacterial counts of wastewater sampling points. (A) Map of Ghana, indicating the location of Tamale (black star), and map of the metropolitan area of Tamale. Sampling locations: TWH, Tamale West Hospital; TCH, Tamale Central Hospital; TTH, Tamale Teaching Hospital; and UWTP, Urban Waste Treatment Plant. Sampling points are displayed by different colored spots in each sampling location (see the legend in the bottom right corner): 1, canalization before the hospital/first treatment pond in UWTP; 2, drainage of the hospital/last treatment pond in UWTP; and 3, canalization after the hospital/drainage of the UWTP. Geographical coordinates of sampling points are indicated in Table S1. An interactive map is available at the Microreact website (https://microreact.org/project/2sBABj7YhfhoLZi8trXvSv). (B) Bacterial counts (CFU/mL) of total bacteria, carbapenem-resistant bacteria, and pan-aminoglycoside-resistant bacteria in wastewater samples from canalizations of Tamale. Total bacteria encompass the entire bacterial population, including carbapenem- and pan-aminoglycoside-resistant bacteria. Sampling locations and points are indicated according to panel A.

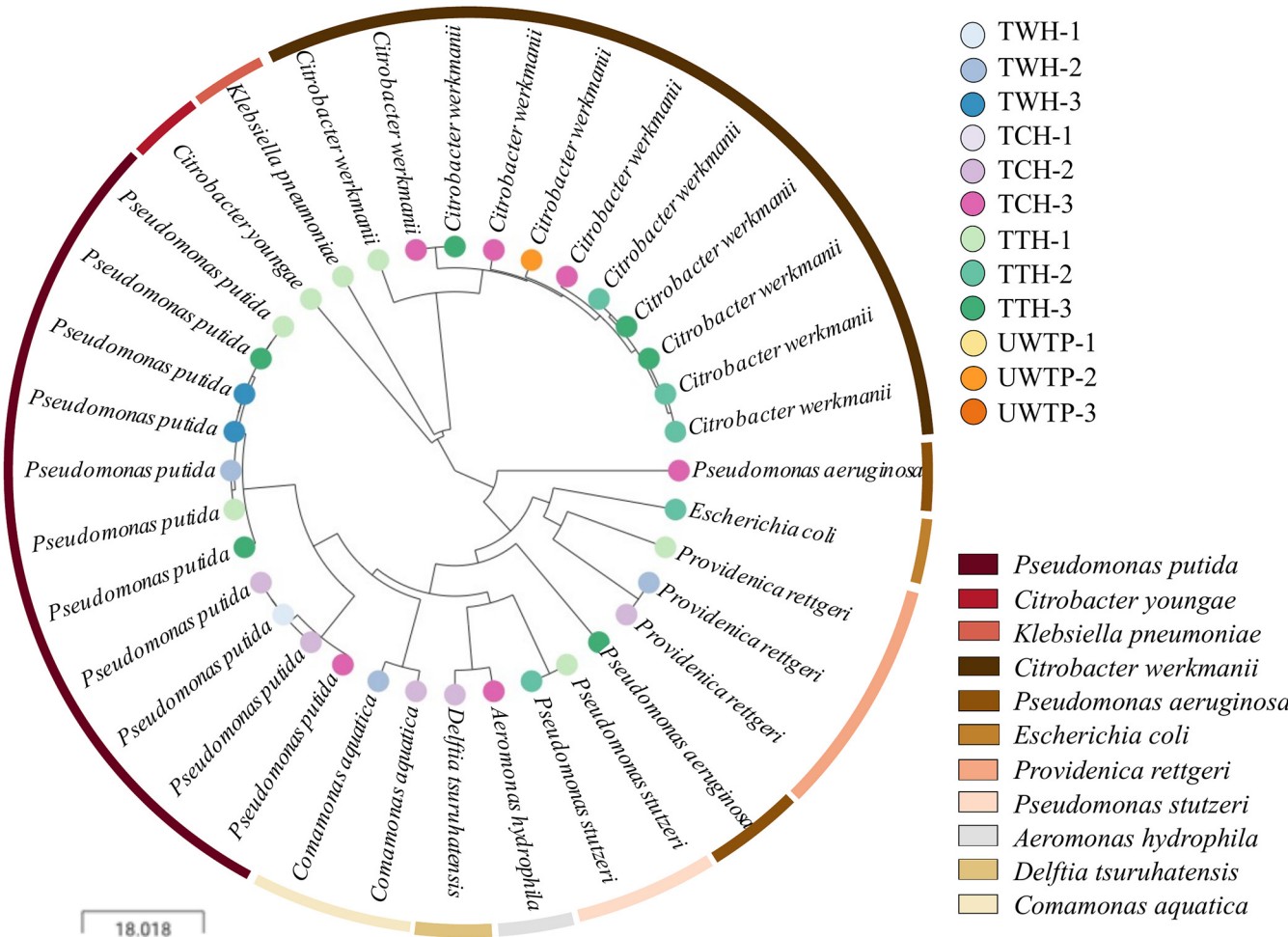

**FIG 2** Tree of carbapenem/pan-aminoglycoside-resistant isolates based on its total resistance gene content. Bacterial species are indicated by leaf labels and colored sections in the outer ring (legend in the bottom right corner of the figure). Sampling points are indicated by colored leaf nodes in the tree (see the legend in the top right corner). Sampling locations: TWH, Tamale West Hospital; TCH, Tamale Central Hospital; TTH, Tamale Teaching Hospital; and UWTP, urban waste treatment plant. Sampling points: 1, canalization before the hospital/first treatment pond in UWTP; 2, drainage of the hospital/last treatment pond in UWTP; and 3, canalization after the hospital/drainage of the UWTP. An interactive visualization, together with the map of Tamale, is available at the Microreact website (https://microreact.org/project/2sBABj7YhfhoLZi8trXvSv).

other β-lactam compounds, all *P. putida* isolates also harbored a variant of the carbenicillinase gene *bla*$_{CARB}$ (Fig. 3A to C). Another uncommon carbapenemase gene, *bla*$_{DIM-1}$, was also identified in *P. putida* isolates, but only isolates belonging to ST125. No plasmid incompatibility group nor potential plasmid marker included in the PlasmidFinder databases was identified in *P. putida* isolates (Fig. 3A to C). Pan-genome analysis of 11 *P. putida* isolates showed the genetic divergence between the two STs identified, comprised by a total of 9,863 gene clusters and a core genome of 1,081 (10.96%), showing the broad and variable genetic collection of this species and, even, intraspeciation. Nevertheless, the core genome among ST125 isolates contained 5,116 out of the total 5,266 gene clusters (97.15%), and similarly, the ST126 core genome was comprised of 5,338 out of 5,683 gene clusters (93.93%), demonstrating the high clonality of *P. putida* circulating in different wastewater canalizations across the region (see Fig. S1A in the supplemental material).

*C. werkmanii* isolates were collected from two of the screened hospitals and the UWTP, mostly in wastewater samples from the drainage of the hospitals and canalizations after the hospitals, probably due to the multiplication effect observed in these environments. All *C. werkmanii* isolates shared a common multidrug resistance profile, only exhibiting susceptibility to the last-line antibiotics colistin and tigecycline. Resistance to carbapenems was a consequence of the carbapenemase gene *bla*$_{NDM-1}$. They also presented other genes conferring resistance to β-lactams, namely, the extended-spectrum β-lactamase (ESBL) gene

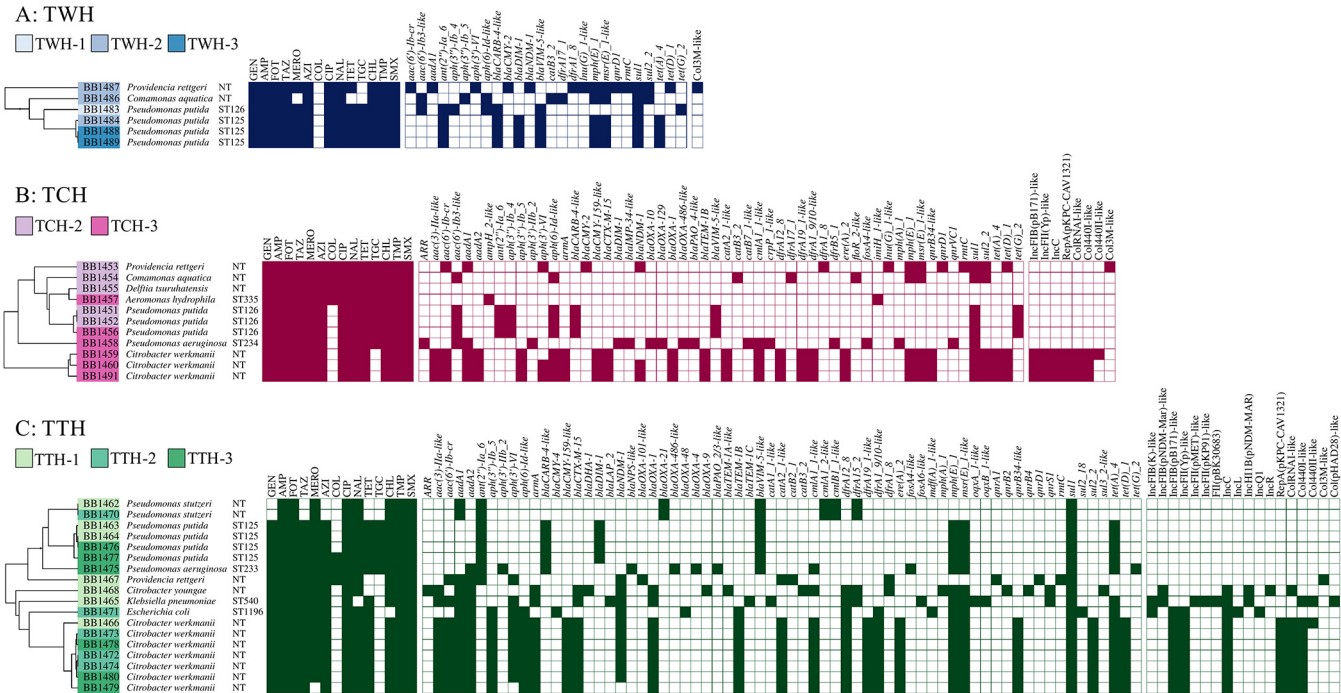

**FIG 3** Carbapenem/pan-aminoglycoside-resistant bacterium data. Trees are constructed based on the similarity between total resistance gene content. The sampling point from where each isolate was recovered is specified by different colored shades in the isolate code. Sampling points: 1, canalization before the hospital; 2, drainage of the hospital; and 3, canalization after the hospital. Bacterial species and sequence types are indicated following the isolates' codes. NT, nontypeable ST. Phenotypic resistance to tested antibiotics is indicated by filled colored squares and phenotypic susceptibility by empty squares. Antibiotics: GEN, gentamicin; AMP, ampicillin; FOT, cefotaxime; TAZ, ceftazidime; MERO, meropenem; CHL, chloramphenicol; TMP, trimethoprim; AZI, azithromycin; COL, colistin; CIP, ciprofloxacin; NAL, nalidixic acid; TET, tetracycline; TGC, tigecycline; and SMX, sulfamethoxazole. The presence of antibiotic resistance genes and plasmid incompatibility groups is indicated by filled colored squares, and the absence of those genetic elements is indicated by empty squares. (A) Tamale West Hospital (TWH). (B) Tamale Central Hospital (TCH). (C) Tamale Teaching Hospital (TTH).

$bla_{CTX-M-15}$, AmpC $\beta$-lactamase gene $bla_{CMY}$, and the oxacillinase gene $bla_{OXA}$. *C. werkmanii* isolates were also pan-aminoglycoside resistant because of the 16S RMTase gene *armA*, detected in all isolates. Plasmid incompatibility groups identified in *C. werkmanii* isolates were numerous and stable between them, including IncC and various IncF plasmid types (Fig. 3B and C). Similar to the *P. putida* situation, the 11 *C. werkmanii* isolates included in the pan-genome analysis showed a close relationship between all of them, displaying a common core genome with 5,604 gene clusters out of a total collection of 5,978 (93.74%), and pointing out the local spread of the same *C. werkmanii* cluster via wastewater (Fig. S1B).

The $bla_{NDM-1}$ gene was also detected in all *P. rettgeri* isolates, which were found in the three hospitals. These isolates exhibited a variable multidrug resistance profile, including a pan-resistant isolate from TCH. Resistance to all clinically relevant aminoglycosides was conferred by a 16S RMTase gene, but the responsible gene in this species was *rmtC*. No other plasmid marker was identified by using the PlasmidFinder databases, apart from a Col3M-like-type plasmid (Fig. 3A to C).

One isolate each of *Comamonas aquatica* and *Delftia tsuruhatensis* exhibited a pan-resistant profile, including a high level of resistance to carbapenems and aminoglycosides. However, no resistance gene included in the ResFinder database nor a similar genetic determinant responsible for this phenotype was identified (Fig. 3B). Further genomic studies are required to uncover the resistance mechanisms in these environmental species.

**Novel and uncommon carbapenemase/$\beta$-lactamase gene variants were disseminated by clonal expansion of environmental *Pseudomonas* spp. and *Citrobacter* spp.** Hybrid assemblies from combined Illumina and Nanopore data and subsequent analyses allowed the characterization of carbapenemase and 16S RMTase genes, including their genomic location and association with MGEs. Two novel genetic variants of the $bla_{VIM}$ carbapenemase gene were identified in *Pseudomonas* spp. Both were related to $bla_{VIM-5}$ and shared a point mutation (321A→G) with no amino acid modification. However, one of these genetic

variants presented another point mutation (523C→G), which produced an amino acid change (Pro175Ala), resulting in a 99.6% amino acid identity with VIM-5 carbapenemase (see Fig. S2A in the supplemental material). This novel variant was named $bla_{VIM-71}$ (following the previously described variants), and it was highly associated with *P. putida* isolates. $bla_{VIM-71}$ was always integrated in the chromosome and was surrounded by a common genetic structure composed by other resistance genes and MGEs, including IS*Pa* variants (Fig. 4A). Furthermore, the genetic variant of $bla_{VIM-5}$ with no amino acid changes was only identified in *Pseudomonas stutzeri* isolates on a novel plasmid type with a RepE origin of replication (pPS-VIM-5). This plasmid showed a maximum nucleotide identity of 69.70% over the 79% of the sequence length with other plasmids characterized in *Pseudomonas* spp. $bla_{VIM-5}$ was integrated in the variable region of the plasmid, flanked by MGEs and other relevant resistance genes, such as $bla_{OXA-21}$ (Fig. 4B).

A novel variant of the carbenicillinase gene $bla_{CARB-4}$ was identified among the resistance genes located close to $bla_{VIM-71}$ in the chromosome of *P. putida* isolates. This variant, named $bla_{CARB-53}$, presented a nucleotide point mutation (388G→T) that resulted in one amino acid variation (Ala130Ser) and displayed a 99.7% identity to the CARB-4 protein (Fig. S2B). Similar to $bla_{VIM-71}$, $bla_{CARB-53}$ was flanked by several MGEs, including the IS*Pa* variants and other ISs, and the GC content of the gene was particularly lower than that in the chromosome (38.4% versus 62.5%) (Fig. 4A). Both resistance genes were also identified in the chromosome of one *P. aeruginosa* isolate, integrated separately but associated with the same flanking IS*Pa* variants found in *P. putida* chromosomes.

One of the two *P. putida* STs identified, ST125, also harbored an uncommon carbapenemase gene, $bla_{DIM-1}$. This gene was carried in a nontypeable plasmid, named pPP-DIM-1, with two RepA origins of replication. The pPP-DIM-1 *repA* sequences shared a high nucleotide identity with a limited number of plasmids previously described in *Pseudomonas* spp. Focusing on the gene, $bla_{DIM-1}$ was integrated in the variable region, flanked by numerous MGEs, Tn*As*-like transposons among them (Fig. 4C).

Regarding *C. werkmanii*, all isolates harbored an unknown variant of the AmpC β-lactamase gene $bla_{CMY}$. This variant differed from $bla_{CMY-159}$ in four nucleotide point mutations (62C→T, 327C→T, 1134G→A, and 1140A→G), one of them producing an amino acid modification (Ala21Val) with respect to $bla_{CMY-159}$, sharing a 99.7% amino acid identity (Fig. S2C). The novel variant was named $bla_{CMY-172}$, following the order of previous described gene variants. This gene was integrated in all *C. werkmanii* chromosomes, sharing a common genetic environment with no related MGEs.

**Environmental *P. rettgeri* and *Citrobacter* spp. as reservoirs of high-risk resistance genes and generators of novel gene-MGE-plasmid associations responsible for antimicrobial resistance dissemination.** Contrary to the aforementioned carbapenemase and β-lactamase genes, the carbapenemase gene $bla_{NDM-1}$ was detected in multiple bacterial species and associated with diverse plasmid structures. In the case of *P. rettgeri* isolates, $bla_{NDM-1}$ was always located in a nontypeable plasmid with a RepB origin of replication. This $bla_{NDM-1}$-carrying plasmid (hereinafter called pPR-NDM-1A) showed a flexible structure in different *P. rettgeri* isolates, some of them presenting an ~100-kb region formed by multiple genes from the conjugation machinery of an IncC plasmid, but no other origin of replication (pPR-NDM-1B) (see Fig. S3A in the supplemental material). Nevertheless, the region surrounding $bla_{NDM-1}$ displayed a conserved genetic structure in *P. rettgeri*, including the closely related $bla_{NDM-1}$ MGEs IS*Aba125* and IS*CR1* and frequently associated resistance genes, such as the AmpC β-lactamase gene $bla_{DHA-1}$ and the 16S RMTase gene *rmtC* (Fig. 5A). The latter was placed together with IS*3000*, but this 16S RMTase gene was not identified in species other than *P. rettgeri*.

The most prevalent $bla_{NDM-1}$-carrying species was *C. werkmanii*. In this case, the main plasmid involved in $bla_{NDM-1}$ dissemination was an IncC-type plasmid broadly identified in *Enterobacteriaceae*. This plasmid, referred to as pCW-NDM-1, displayed an ~162-kb structure among *C. werkmanii* isolates, including the $bla_{NDM-1}$ flanking region with IS*Aba125* and IS*CR1*, shared with *P. rettgeri*. The 16S RMTase gene associated with $bla_{NDM-1}$ in pCW-NDM-1 was *armA*. The genetic context of *armA* was formed by the typical Tn*1548* configuration, including the conserved MGEs IS*Ec28* and IS*Ec29* (Fig. 5B). The pCW-NDM-1-like plasmid found in

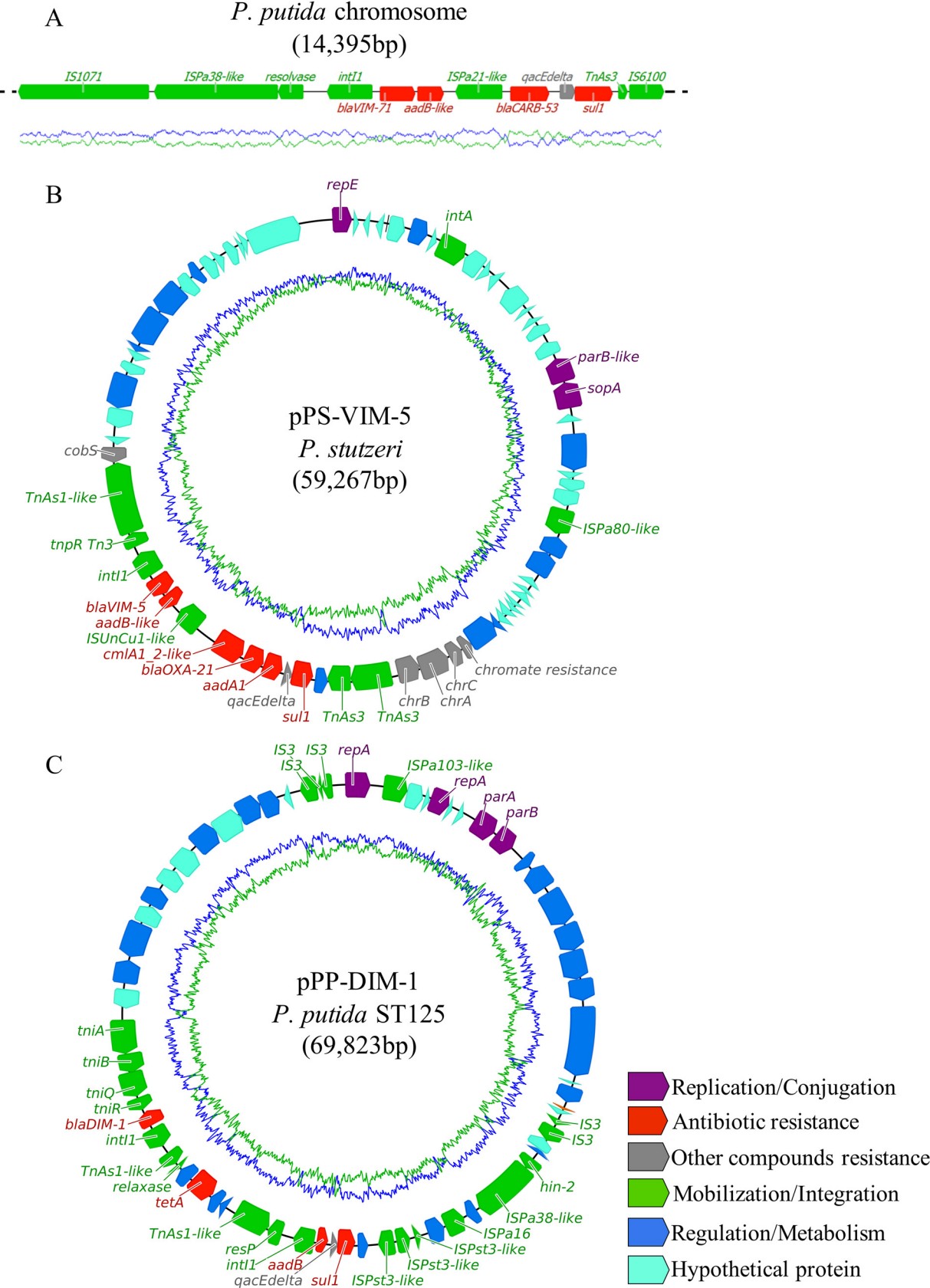

**FIG 4** Genetic organization of carbapenemase genes strongly associated with specific mobile elements and bacterial species. Colored arrows indicate the functional group for each gene (bottom legend) and sequence direction. Manually curated genes are shown by labels (except for the Regulation/Metabolism and Hypothetical Protein groups). (A) *P. putida* chromosome region surrounding the novel genetic variants $bla_{VIM-71}$ and $bla_{CARB-53}$. (B) *P. stutzeri* novel plasmid pPS-VIM-5 harboring $bla_{VIM-5}$. (C) *P. putida* ST125 novel plasmid harboring $bla_{DIM-1}$.

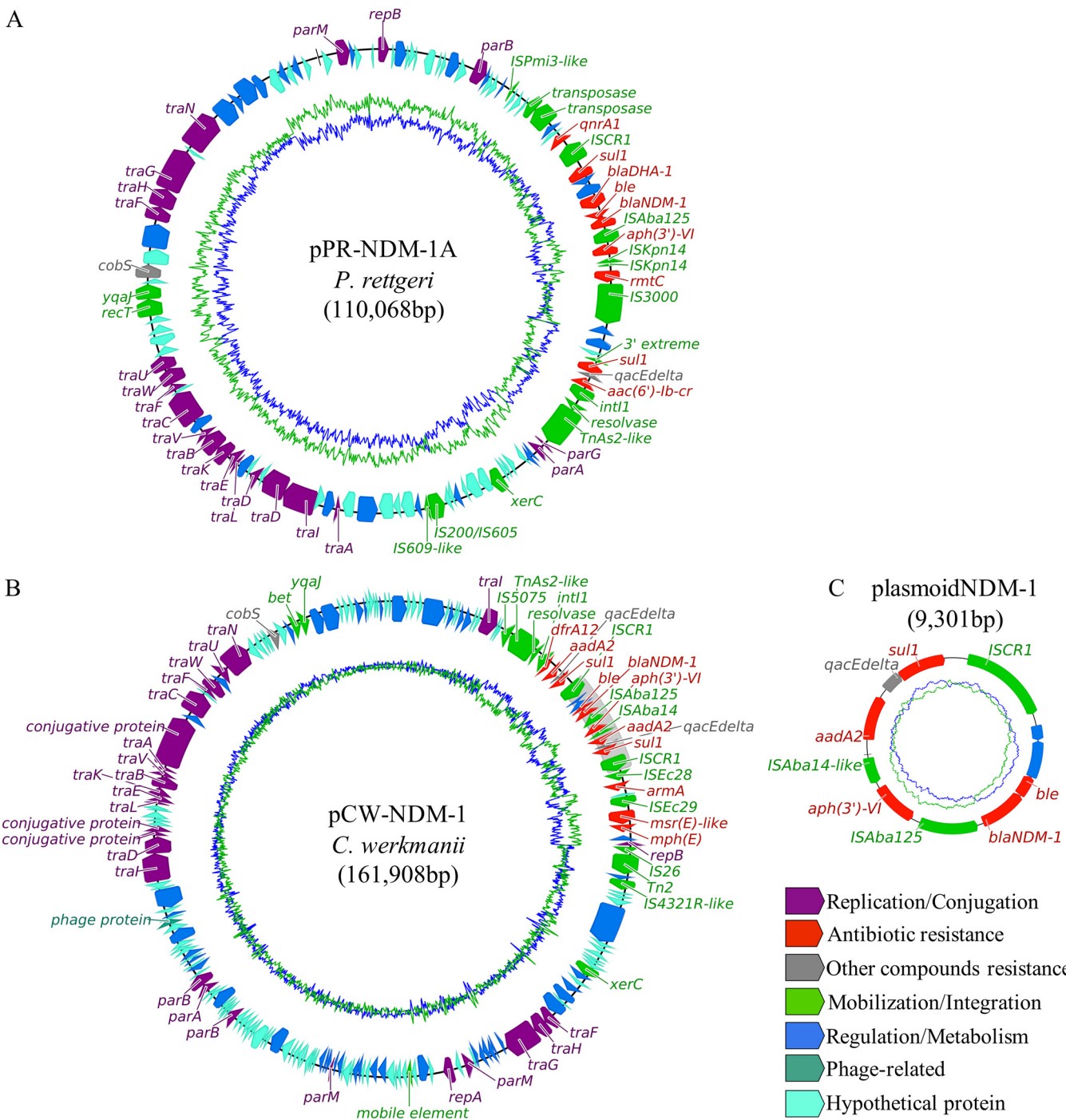

**FIG 5** Genetic organization of *bla*<sub>NDM-1</sub> and its multiple carrying structures. Colored arrows indicate the functional group for each gene (bottom legend) and sequence direction. Manually curated genes are shown by labels (except for the Regulation/Metabolism and Hypothetical Protein groups). (A) *P. rettgeri* pPrY2001-like plasmid pPR-NDM-1A harboring *bla*<sub>NDM-1</sub> and *rmtC*. (B) *C. werkmanii* IncC plasmid pCW-NDM-1 harboring *bla*<sub>NDM-1</sub> and *armA*. Gray shading highlights the plasmoid element when it is integrated in the IncC plasmid. (C) *Enterobacteriaceae* plasmoid element harboring *bla*<sub>NDM-1</sub>.

*Citrobacter youngae* (pCY-NDM-1) possessed an additional ~90-kb region placed by a Tn*2* structure. This region carried a collection of antimicrobial resistance genes, including the oxacillinase gene *bla*<sub>OXA-9</sub> and a novel putative metallo-β-lactamase gene, as well as metal resistance genes, such as those conferring resistance to tellurite (Fig. S3B). Additionally, in two *C. werkmanii* isolates, one *Escherichia coli* isolate, and one *Klebsiella pneumoniae* isolate, the gene *bla*<sub>NDM-1</sub> was located in a circular extrachromosomal genetic structure of 9.3 kb. This structure seemed to originate from pCW-NDM-1 due to the split generated by IS*CR1*

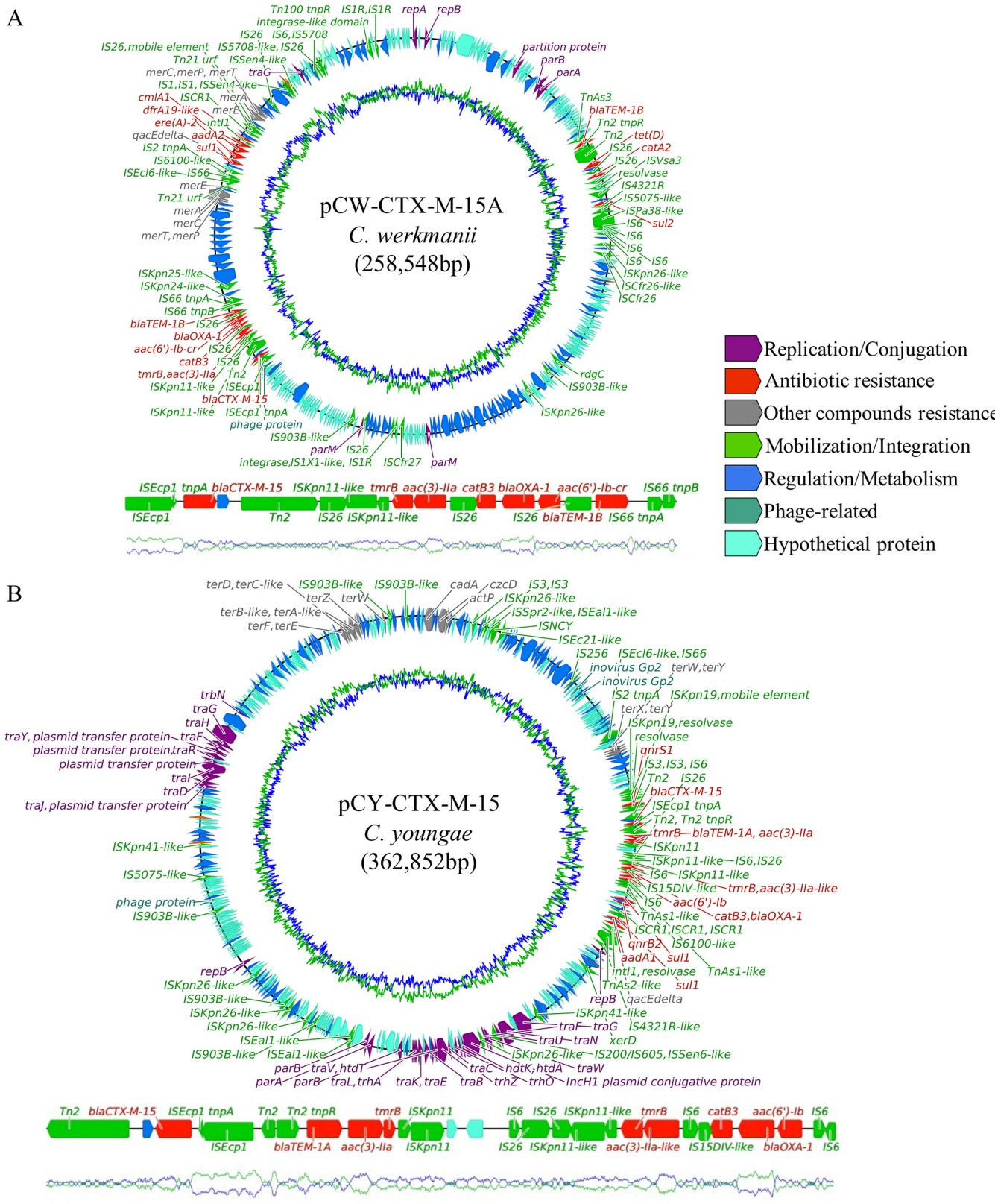

**FIG 6** Genetic organization of diverse plasmid structures harboring $bla_{CTX-M-15}$ and $bla_{OXA-1}$. Colored arrows indicate the functional group for each gene (bottom legend) and sequence direction. Manually curated genes are shown by labels (except for the Regulation/Metabolism and Hypothetical Protein groups). Closer visualizations of the region surrounding resistance genes in the plasmid are displayed at the bottom of each gene. (A) *C. werkmanii* pKPC-CAV1321-like plasmid pCW-CTX-M-15A. (B) *C. youngae* IncHI1B/IncFIB (pNDM-MAR-like) plasmid pCY-CTX-M-15.

flanking $bla_{NDM-1}$ and other resistance genes (Fig. 5C). Furthermore, this "plasmoid" structure presented an average copy number of ~3, compared to the one copy of the IncC plasmid cohabiting in the same cell, according to the comparison of their mean short-read coverage between them and with respect to the bacterial chromosome.

The ESBL gene $bla_{CTX-M-15}$ was closely linked to the oxacillinase gene $bla_{OXA-1}$, both of which were carried in a plasmid belonging to the pKPC-CAV1321 type. This plasmid, referred as pCW-CTX-M-15A, was identified in most *C. werkmanii* isolates and displayed a conserved configuration, including the $bla_{CTX-M-15}$/$bla_{OXA-1}$ region. Among the MGEs responsible for this region's plasticity, IS*Ecp1* was especially related to $bla_{CTX-M-15}$, whereas $bla_{OXA-1}$ was surrounded by multiple IS*26* MGEs (Fig. 6A). Moreover, $bla_{CTX-M-15}$ was found in all *C. werkmanii* chromosomes, inserted together with the IS*Ecp1*.

*C. youngae* also harbored $bla_{CTX-M-15}$ and $bla_{OXA-1}$, but in an alternative plasmid structure with two different replicon types, IncHI1B and IncFIB. The flanking regions of $bla_{CTX-M-15}$ and $bla_{OXA-1}$ in this plasmid, referred to as pCY-CTX-M-15, presented similar MGE gene organizations to those found in pCW-CTX-M-15A, with IS*Ecp1*-$bla_{CTX-M-15}$ and IS*26*-$bla_{OXA-1}$ modules, but with certain rearrangements caused by IS*26* (Fig. 6B). Furthermore, another copy of $bla_{OXA-1}$ was also identified and carried by an IncR plasmid (pCY-OXA-1) in *C. youngae*, and the gene was associated with MGEs other than IS*26*.

**Plasmid fusion allowed plasmid adaptation to diverse ecological niches and configuration of novel multireplicon structures.** In *C. werkmanii*, the cohabitation of pCW-NDM-1 and pCW-CTX-M-15A led to the complete plasmid fusion of a multireplicon RepA/IncC plasmid via the Tn*2* structure present in both parental plasmids. The resulting plasmid (pCW-CTX-M-15B) carried $bla_{CTX-M-15}$ and $bla_{OXA-1}$ from pCW-CTX-M-15A and *armA* from pCW-NDM-1, among other resistance genes. However, $bla_{NDM-1}$ was not present in the plasmid, due to the absence of the gene in the initial pCW-NDM-1 or the previous split of the gene via IS*CR1* mobilization (see Fig. S4A in the supplemental material).

The fusion of pCY-CTX-M-15 and pCW-NDM-1 led to the formation of an IncHI1B/IncFIB/IncC hybrid plasmid via the Tn*As2*-like element present in both plasmids. This structure (pKP-CTX-M-15) identified in *K. pneumoniae* also combined $bla_{CTX-M-15}$, $bla_{OXA-1}$, and *armA* from its parental plasmids, among other genes associated with resistance to other antibiotic classes, such as quinolones and macrolides (Fig. S4B). The same isolate carried the "plasmoid" structure caused by $bla_{NDM-1}$-IS*CR1* module mobilization from the preliminary pCW-NDM-1.

## DISCUSSION

Antimicrobial resistance dynamics in LMICs result from the integration of particular geographic, cultural and socioeconomic factors different than those in regions with suitable community infrastructures and health systems (3). This is especially significant in the wastewater scenario, where the flux and collection of genetic and bacterial elements from diverse sources favor the emergence, spread, and association of multiple resistance mechanisms, including those conferring resistance to last-resort antibiotics, such as carbapenems and aminoglycosides (1). In the metropolitan area of Tamale, the increase of MDR bacteria levels from hospital and UWTP canalizations, compared with those observed before the hospitals, revealed the multiplier and gatherer effect of these environments, as previously reported (5). Resistant bacteria either could be present in wastewater prior to the hospitals/UWTP at low levels or could be native to these environments, from where they could potentially be released into urban community canalizations. The high levels of MDR bacteria found in all sampling points of TTH could be explained by the wastewater flux across the canalizations in Tamale, which followed a northwest-southeast path. Consequently, this wastewater flux generated a parallel bacterial flux, accumulating bacteria in a progressive fashion from community and hospital settings throughout the region. TWH and TCH were located closer to the northwest-central area of Tamale, so the bacterial input from previous wastewater canalization had a minor load of resistant bacteria compared to TTH, which is a larger hospital located closer to the southeast area of the region, and therefore has a higher input of resistant bacteria from previous canalizations (Fig. 1A) (see Microreact at https://microreact.org/project/2sBABj7YhfhoLZi8trXvSv). The flux of MDR bacteria through both

community and hospital wastewater canalizations of the region was confirmed by the high clonality of *P. putida* and *C. werkmanii* isolates from all canalization types and sampling locations, which together comprised 61% of the resistant bacterial diversity circulating in the region. Even the resistant *C. werkmanii* isolate found in UWTP ponds belonged to the same cluster identified in the urban and hospital canalizations of the central area of Tamale, showing the long-range spread of these bacteria through wastewater.

The dissemination of numerous rare carbapenemase/$\beta$-lactamase gene variants associated with specific environmental bacterial species pointed out the large clonal and endemic expansion of these gene-species associations in wastewater of the region and, thus, in the community bacterial populations (16). Some of these variants, including the novel $bla_{VIM-71}$ and $bla_{CARB-53}$ genes identified in *P. putida* isolates, were integrated in the chromosome and surrounded by MGEs, which could potentially be responsible for genetic mobilization and be involved in further dissemination events to other bacterial hosts, as reported for other related gene variants (17, 18). In fact, both genes were identified in the chromosome of one *P. aeruginosa* isolate surrounded by the same MGEs detected in *P. putida*, confirming the feasibility of interspecies dissemination. Moreover, the GC content discrepancy between $bla_{CARB-53}$ and the *P. putida* chromosome denoted a different bacterial origin of the gene and a relatively recent integration in this species' chromosome. Other carbapenemase resistance genes in *Pseudomonas* spp. were associated with very specific plasmids, disseminated by plasmid-species expansion. This is the case of $bla_{VIM-5}$ carried by a novel *Pseudomonas* plasmid type in *P. stutzeri* and $bla_{DIM-1}$ linked to a nontypeable *Pseudomonas* plasmid in *P. putida* ST125 isolates. These genes, also flanked by diverse MGEs, could be potentially transferred and mobilized among closely related bacteria via plasmid dissemination and transposon integration, as described for similar gene-MGE associations (19, 20). Contrarily, the absence of MGEs surrounding the novel variant $bla_{CMY-172}$, identified in the chromosome of all *C. werkmanii* isolates, pointed to this species as the possible bacterial origin of the gene or the existence of a close and long gene-species association with a low risk of gene mobilization.

Concurrently to the genetic dissemination caused by clonal expansion of bacterial hosts, other carbapenemase genes were associated with multiple MGEs, demonstrating a high competence to be mobilized, transferred, and coselected. In fact, the genetic dissemination via horizontal gene transfer is one of the main AMR drivers in *Enterobacteriaceae* species inhabiting wastewater environments (21). This is the case for $bla_{NDM-1}$, identified in diverse plasmid structures but generally sharing a common gene-flanking region. The gene was carried by a species-specific plasmid and was associated with the 16S RMTase gene *rmtC* in *P. rettgeri*, which could be the initial reservoir of this gene, as previously elucidated in other studies (22). pPR-NDM-1A/B possessed an origin of replication highly similar to pPrY2001-like plasmids identified in *P. rettgeri* and *Proteus mirabilis* (23). The presence of a large region from the conjugation machinery of IncC plasmids in pPR-NDM-1B pointed at the earlier contact of these two plasmid structures. Besides, the integration of $bla_{NDM-1}$ in the IncC plasmid pCW-NDM-1 harbored by all *C. werkmanii* isolates, which shared a high nucleotide identity with the IncC region found in pPR-NDM-1B, could have originated in *P. rettgeri*. This integration resulted in the combination of $bla_{NDM-1}$ with another 16S RMTase gene, *armA*, in the same plasmid structure, favoring a different high-risk resistance gene association.

In addition to the commonly $bla_{NDM-1}$-linked IS*Aba125* (24), the gene was flanked by IS*CR1* in pCW-NDM-1 found in *Citrobacter* species isolates. This IS*91* family element can be split and mobilized, together with adjacent genes, by rolling circle replication due to the *ori* IS present in its sequence, generating nonreplicative circularized molecules that can be eventually integrated in other genomic locations (25, 26). The 9.3-kb structure carrying $bla_{NDM-1}$ caused by the IS*CR1* split was identified in multiple *Enterobacteriaceae* species from different wastewater canalizations and in a higher copy number than pCW-NDM-1. This evidence suggests the existence of an autonomous replication machinery in this structure, called a "plasmoid," that could allow its persistence in different bacterial species, from where it can be mobilized, disseminating the associated resistance genes. Further studies should be performed to understand the biological functions of "plasmoid" structures.

Similar to the $bla_{NDM-1}$ scenario, $bla_{CTX-M-15}$ and $bla_{OXA-1}$ were disposed in specific MGE

mSystems®

resistance gene modules that could be eventually transferred either as a whole or as independent units. $bla_{CTX-M-15}$ was linked to the typical IS$Ecp1$ (27) and broadly distributed in *C. werkmanii* isolates, an association recently described in Africa (7). It was integrated in both the chromosome and pCW-CTX-M-15A plasmid, which is related to other plasmids harbored by diverse *Citrobacter* and *Klebsiella* spp. (28). This MGE resistance gene module was likely transferred from pCW-CTX-M-15A to the chromosome, demonstrating the capacity of mobilization and fixation of these acquired genes in the bacterial population. $bla_{OXA-1}$ was also located in pCW-CTX-M-15A close to $bla_{CTX-M-15}$, but in an IS$26$-rich region, which are hot spots for integration of other IS$26$-carrying modules and, therefore, for resistance gene collection (29). Likewise, $bla_{CTX-M-15}$ and $bla_{OXA-1}$ modules were identified in a different species, *C. youngae*, and plasmid, pCY-CTX-M-15, an IncHI1B/IncFIB double-replicon plasmid highly similar to pNDM-MAR, a plasmid broadly described in *K. pneumoniae* and linked to the dissemination of multiple carbapenemase genes in clinical and environmental isolates of this species (30). This points out the possible transference of these resistance genes between different plasmids, which could favor the dissemination of $bla_{CTX-M-15}$ and $bla_{OXA-1}$ to new bacterial hosts and niches. Moreover, the $bla_{OXA-1}$-carrying IncR plasmid, cohabiting with pCY-CTX-M-15 but lacking the $bla_{OXA-1}$-flanking IS$26$ copies, suggests multiple routes of gene dissemination via parallel plasmid acquisitions.

The circulation of different plasmids among *Enterobacteriaceae* isolates caused the emergence of novel cointegrative plasmid structures and generated novel resistance gene associations. The main plasmid responsible for these multiplasmid rearrangements was the IncC plasmid pCW-NDM-1, which showed a high interspecies promiscuity, as observed in other IncC plasmids (31). pCW-NDM-1 showed a mosaic structure, exhibiting a high capacity to gain, lose, and rearrange from small genetic regions to large plasmid modules and even to be completely cointegrated with other plasmid structures. The plasmid fusion of pCW-NDM-1 and pCW-CTX-M-15A generated a RepA/IncC plasmid carrying $bla_{CTX-M-15}$, $bla_{OXA-1}$, and *armA*, creating a new carbapenemase-16S RMTase gene association. The convergence of pCW-NDM-1 and pCY-CTX-M-15 in some *Enterobacteriaceae* species, likely in *C. youngae*, was the origin of the IncHI1B/IncFIB/IncC multireplicon plasmid, also containing $bla_{CTX-M-15}$, $bla_{OXA-1}$, and *armA*. Most importantly, these multireplicon structures merged different replicative and conjugative machineries, which potentially increases their capacity of dissemination to new bacterial hosts (32) and ecological niches and, therefore, the capacity of dissemination of their related resistance genes.

In conclusion, wastewater of Tamale constituted an ecological niche where antimicrobial resistance genes, MGEs, and bacteria disseminated with no restrictions across all canalizations. Community wastewater and environmental bacteria were probably the source of genetic diversity, whereas hospital environments selected and amplified successful genetic associations, which returned to community wastewater. Resistance gene dissemination followed diverse and simultaneous strategies: clonal bacterial expansion, MGE mobilization and integration, and plasmid transference and fusion. This "nested Russian doll" model of AMR dynamics has been previously described as the driving force for the emergence of MDR bacteria in hospital patients and health care environments (22, 28, 33). Here, we show the emergence and dissemination of these MDR bacteria across the population of a whole region. Based on data from the present study, public health measures must be established to control the impact of already spreading high-risk MDR bacteria and to avoid the emergence of future ones.

## MATERIALS AND METHODS

**Sampling and bacterial isolation.** Twenty-four wastewater samples from the different locations mentioned were collected in August 2017 in the metropolitan area of Tamale, Ghana. Based on a previous evaluation of the wastewater canalizations, the sampling was designed in order to determine the relationship between MDR bacteria from hospital wastewater and urban wastewater. Four sampling locations were selected, including three hospitals (Tamale West Hospital [TWH], Tamale Central Hospital [TCH], and Tamale Teaching Hospital [TTH]) and one urban waste treatment plant (UWTP). In each of the four sampling locations, three duplicate samples of 50 mL were obtained from three different points: the wastewater urban canalization before the hospital (the first treatment pond in the case of the UWTP), the drainage of the hospital onto the urban canalization (the last treatment pond of the UWTP), and the wastewater urban canalization after the hospital (the point where treated water is discharged to the environment after the UWTP). The three sampling points within the same sampling location were at least 100 m apart from each other to ensure a sufficient distance between different wastewater canalizations (Table S1). Therefore, a total of 24 wastewater samples were collected (4 sampling locations × 3 sampling points × 2 wastewater samples).

To determine the general bacterial population from each sampling point, wastewater samples were serially diluted 10-fold in phosphate-buffered saline (PBS), and 100 $\mu$L from each dilution was plated onto MacConkey agar plates with no antibiotic pressure (Oxoid, Ltd., Basingstoke, Hampshire, United Kingdom), calculating the number of CFU/mL per sample. For counting and isolation of pan-aminoglycoside-resistant bacteria, 100 $\mu$L from each of the aforementioned dilutions was plated onto MacConkey agar plates supplemented with 200 mg/L gentamicin and 200 mg/L amikacin (Sigma-Aldrich, Inc., Saint Louis, MO, USA) (15). Likewise, carbapenem-resistant bacteria were counted and isolated by plating 100 $\mu$L from each 10-fold dilution onto MacConkey agar plates supplemented with 8 mg/L imipenem (Sigma-Aldrich, Inc.) (34, 35). Up to three different colonies from each antibiotic-supplemented medium, when available, were selected from each sample to perform subsequent analyses.

**Bacterial identification, antibiotic susceptibility testing, and detection of 16S RMTase and carbapenemase genes by PCR.** Bacterial species determinations were carried out by MALDI-TOF mass spectrometry in Centro de Vigilancia Sanitaria Veterinaria (VISAVET Health Surveillance Centre, Madrid, Spain), with a cutoff value of ≥2.3 for accurate species identification. MICs of different antimicrobial classes were evaluated by a broth microdilution method using commercial Sensititre EUVSEC plates (Trek Diagnostics, Inc., Westlake, OH, USA) following the manufacturer's specifications. The results were interpreted according to the European Committee on Antimicrobial Susceptibility Testing (EUCAST) guidelines (34), and for antibiotics and bacterial species with no established breakpoints in EUCAST documents, the Clinical and Laboratory Standard Institute (CLSI) guidelines were applied (36). Intrinsic resistance levels related to the different bacterial species identified were also assessed following the EUCAST advice on intrinsic resistance and exceptional phenotypes (37). If no data on clinical resistance breakpoints were available, MICs were evaluated according to the EUCAST epidemiological cutoff values (ECOFFs) (38).

For isolates highly resistant to aminoglycosides, all known 16S RMTase genes (*armA*, *rmtA*, *rmtB*, *rmtC*, *rmtD*, *rmtE*, *rmtF*, *rmtG*, *rmtH*, and *npmA*) were screened by PCR as previously described (39). In the case of carbapenemase-resistant isolates, PCRs were performed following previous studies in order to cover a broad variety of the most prevalent carbapenemase genes: $bla_{VIM}$ (variants 1 and 2), $bla_{IMP}$ (all variants except 9, 16, 18, 22, and 25), $bla_{KPC}$ (variants 1 to 5), $bla_{OXA-48}$-like, $bla_{GES}$ (variants 1 to 9 and 11) (40), $bla_{NDM-1}$ (41), and $bla_{CTX-M}$ (all variants) (42).

**Whole-genome sequencing by Nanopore, data processing, and long-read assembly and analysis.** Selected isolates belonging to different bacterial species resistant to either aminoglycosides or carbapenems from each sample, excluding intrinsic carbapenem-resistant species, were selected for Nanopore whole-genome sequencing (WGS) using the MinION device (Oxford Nanopore Technologies, Ltd., Oxford Science Park, Oxford, United Kingdom). Genomic DNA extraction and purification were performed by the Wizard Genomic DNA purification kit (Promega Corp., Madison, WI, USA), following the "Isolating Genomic DNA from Gram Negative Bacteria" protocol. Subsequently, DNA quality and concentration were measured by the NanoDrop (Thermo Fisher, Inc., Waltham, MA, USA) and Qubit (Invitrogen Corp., Carlsbad, CA, USA) platforms. Genomic libraries were prepared following the "1D Native Barcoding Genomic DNA" protocol, using the EXP-NBD104 and SQK-LSK109 kits (Oxford Nanopore Technologies, Ltd.), and sequencing was run in a FLO-MIN106 flow cell. Downstream analysis was performed as follows: sequencing reads were base called with MinKNOW software (Oxford Nanopore Technologies, Ltd.), the demultiplexing process was developed via the Fastq barcoding workflow of the Epi2Me interface (Metrichor, Ltd., Oxford Science Park, Oxford, United Kingdom), and trimming of adaptors and barcodes from the reads was performed by Porechop version 0.2.3 (43), obtaining a mean of 267-fold coverage (minimum, 51-fold; maximum 756-fold). Raw Nanopore sequence data were deposited in the European Nucleotide Archive (ENA) (44) under project no. PRJEB38443, and individual accession numbers are indicated in Table S2 in the supplemental material.

Assemblies with long-read sequences were created with Flye version 2.4.2 (45), and their qualities were assessed with QUAST version 5.0.2 (46), obtaining $N_{50}$ values between 4,054,225 and 7,019,734 bp. Genomic structures were visualized with Bandage version 0.8.1 (47) and analyzed using NCBI BLAST+ version 2.9.0 (48) and applying the ResFinder (49) and PlasmidFinder (50) databases, in order to determine the preliminary resistance gene content (especially 16S RMTase/carbapenemase genes) and the plasmid content, respectively. The long-read assemblies allowed for the determination of the genomic location of the different preliminary resistance genes detected—either chromosomic or plasmidic—and the association between them.

**Whole-genome sequencing by Illumina, data processing, and short-read *de novo* assembly.** High-throughput genome sequencing of all aminoglycoside/carbapenem-resistant isolates was carried out in the Instituto Tecnológico Agrario de Castilla y León (ITACYL). Briefly, 300-bp paired-end read libraries were prepared using the Nextera XT kit and sequenced on a MiSeq platform (Illumina, Inc., San Diego, CA, USA), obtaining a mean of 70-fold coverage (minimum, 15-fold; maximum, 147-fold). Short-read sequences were processed for subsequent analysis by checking their sequencing quality with FastQC version 0.11.8 (51) and trimming end nucleotides with low quality using Trimmomatic version 0.39 (52). Raw Illumina sequence data were deposited under the project no. PRJEB38442 in the ENA, and individual accession numbers are indicated in Table S3 in the supplemental material.

*De novo* assembly of all Illumina-sequenced isolates was carried out with SPAdes version 3.14.0 (53), using paired-end reads and auto k-mer size estimations. Assembly qualities were assessed with QUAST, with $N_{50}$ values between 19,927 and 612,994. Contigs conforming each assembly were annotated by Prokka version 1.14.5 (54). Short-read assembly data were submitted to ENA under the project no. PRJEB38442, and genome accession numbers are indicated in Table S4 in the supplemental material.

**Sequence type, resistome, and plasmidome analyses.** Bacterial species identification was confirmed via Kraken version 2.0.8 (55) and the reference library for bacterial taxonomy from the National Center for Biotechnology Information (NCBI) database (56), using the trimmed short reads obtained from Illumina sequencing. The sequence types (STs) of all aminoglycoside/carbapenem-resistant isolates

were characterized by multilocus sequence typing (MLST) based on the short-read assemblies using the MLSTcheck tool version 2.1.1706216 (57) for all bacterial species with available allele schemes in PubMLST databases (58). The total resistance gene content and plasmid content by incompatibility group classification were identified following two different approaches: using Illumina trimmed short reads with ARIBA version 2.14.4 (59) and SPAdes short-read assemblies with ABRicate version 0.9.8 (60), applying ResFinder and PlasmidFinder databases in both cases. The sampling point, bacterial species, phenotypic resistance profile, resistance gene content, and plasmid content data were integrated for each sampling location, ordered according to resistance gene profile similarity from ARIBA, and visualized with iTOL version 5.5 (61). Moreover, the resistance gene profile tree was also combined with the isolates' metadata (bacterial species, sampling point, and isolation date) to generate an interactive spatiotemporal visualization with the Microreact web tool (62).

**Carbapenemase/$\beta$-lactamase gene analysis.** Identification of novel carbapenemase/$\beta$-lactamase gene variants was conducted by extracting the sequences from the assemblies and comparing them with the closest gene variants of the same gene collected from the NCBI database. Then, the sequences were aligned by MAFFT version 7.407 (63), using the high-accuracy approach and incorporating the local pairwise alignment information. Furthermore, nucleotide sequences were translated into amino acid sequences using Geneious version 8.1.9 (64), and the resulting sequences were used as input to align with MAFFT in order to detect nonsynonymous variations at the protein level. Both nucleotide and amino acid sequence alignments were visualized with Geneious.

**Pan-genome and core and accessory genome analyses.** The pan-genome study was developed for specific bacterial species when the same ST/bacterial species was found in at least two of the different sampling locations in order to assess clonal dissemination across the region, using the program Roary version 3.12.0 (65). Briefly, annotated contigs from Prokka were used as input to obtain gene families belonging to both core and accessory genomes. Single nucleotide polymorphisms (SNPs) were subsequently filtered from shared core genome gene families via the SNP-sites tool version 2.5.1 (66), and variable positions were applied to generate a core genome SNP tree using RAxML version 8.2.8 (67) with 100 bootstrap iterations. Visualization and editing of the tree were performed with FigTree version 1.4.3 (68), and graphical representation of the pan-genome analysis was created with Phandango version 1.3.0 (69), integrating the core genome SNP tree together with the gene presence/absence matrix obtained from Roary pipeline.

**Long-short-read hybrid assembly and genomic analysis.** Nanopore sequencing data and Illumina sequencing data were combined to performed short-long-read hybrid assemblies by using Unicycler version 0.4.7 (70), showing $N_{50}$ values ranging between 2,060,458 and 6,126,880. Complete genomes were evaluated with Bandage and annotated with Prokka. Chromosomes and plasmids carrying targeted resistance genes were located and identified by NCBI BLAST+ with the ResFinder and PlasmidFinder databases, as described in previous sections, in order to detect the resistance gene content and the presence of plasmid incompatibility group sequences, respectively. Subsequently, chromosome and plasmid structures were extracted independently from each annotated genome to perform further analysis. Hybrid assemblies were submitted to ENA under project no. PRJEB38443, and genome accession numbers are indicated in Table S5 in the supplemental material.

**Analysis of 16S RMTase and carbapenemase gene-carrying genomic structures.** Closed plasmid structures from hybrid assemblies carrying either 16S RMTase genes, carbapenemase genes, or both were manually curated from annotated sequences by searching closely related nucleotide sequences in the NCBI database. For those genes integrated in the chromosome, the genetic environment was extracted and equally curated to identify adjacent genes and possible MGEs. Plasmid structures with the same origin of replication or resistance gene content and resistance gene chromosomic environments were aligned by Progressive Mauve version 2.3.1 (71), analyzing the nucleotide identity between sequences and obtaining the common plasmid/chromosome structures, which were graphically visualized using Geneious. The copy number of particular 16S RMTase/carbapenemase gene-carrying structures was estimated from the comparison of the mean coverage, calculated by mapping the quality control (QC) filtered and trimmed short reads against the closed genome, of specific genetic signatures and the whole length of these structures and the bacterial chromosome. In addition, the sequences of most related plasmids sharing an identical or similar origin of replication and plasmid backbone to those detected in the present study were collected from the NCBI database and aligned by Mauve, in order to identify common plasmid structures involved in antimicrobial resistance dissemination. Annotated and curated plasmid structures were submitted to ENA under project no. PRJEB38443, and analysis accession numbers are indicated in Table S6 in the supplemental material.

**Data availability.** All raw and assembled sequence data generated during the current study are available in the European Nucleotide Archive (ENA) at EMBL-EBI under the umbrella project no. PRJEB39000 and are properly specified within the article and the supplemental material.

## SUPPLEMENTAL MATERIAL

Supplemental material is available online only.

**FIG S1**, TIF file, 1.6 MB.

**FIG S2**, TIF file, 2 MB.

**FIG S3**, TIF file, 1.2 MB.

**FIG S4**, TIF file, 2.3 MB.

**TABLE S1**, DOCX file, 0.02 MB.

**TABLE S2**, DOCX file, 0.02 MB.

**TABLE S3**, DOCX file, 0.02 MB.
**TABLE S4**, DOCX file, 0.02 MB.
**TABLE S5**, DOCX file, 0.02 MB.
**TABLE S6**, DOCX file, 0.02 MB.

## ACKNOWLEDGMENTS

We acknowledge Sophia David and David Aanensen from the Centre for Genomic Pathogen Surveillance (CGPS) at the Wellcome Sanger Institute for scientific advice and technical support for the study. We thank Almudena Casamayor from the Microbial Identification and Characterization Unit at the VISAVET Health Surveillance Centre for her contribution in the bacterial identification by MALDI-TOF mass spectrometry.

The work was supported by the Spanish Ministry of Economy and Competitiveness (MINECO BES-2015-073164) and the European Union's Horizon 2020 Research and Innovation Program (grant 773830, OH-EJP-H2020-JRP-AMR-2-WORLDCOM). The funders had no role in study design, data collection, or analysis execution and interpretation.

Author contributions: J.F.D.-B., C.K.S.S., and B.G.-Z. designed the study. C.V.A. and C.K.S.S. collected the samples and performed preliminary screening for carbapenem/pan-aminoglycoside-resistant bacteria. J.F.D.-B., C.V.A., E.M.R., and N.M.S. performed bacterial identification, antimicrobial susceptibility testing, and detection of resistance genes by PCR. J.F.D.-B., E.M.R., C.S.B., and N.M.T. carried out Nanopore sequencing. J.F.D.-B. designed and performed genomic analyses with Illumina and Nanopore data, including genome assemblies, MLST identification, resistome and plasmidome analyses, pan-genome analyses, and structural genomic analyses. C.S.B. cooperated to perform genomic analyses. B.G.-Z. supervised the entire work. J.F.D.-B. and B.G.-Z. wrote the manuscript. All authors contributed to manuscript revision and editing.

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
