## [Reviewer comments · mSystems]

Dissemination routes of carbapenem and pan-aminoglycoside resistance mechanisms in hospital and urban wastewater canalizations of Ghana

Jose Delgado-Blas, Cecilia Valenzuela Agüi, Elena Marín Rodríguez, Carlos Serna, Natalia Montero, Courage Setsoafia Saba, and Bruno Gonzalez-Zorn

Corresponding Author(s): Bruno Gonzalez-Zorn, Universidad Complutense de Madrid

Review Timeline:

Submission Date:	August 17, 2021
Editorial Decision:	October 7, 2021
Revision Received:	December 6, 2021
Accepted:	January 2, 2022

Editor: Rup Lal

Reviewer(s): Disclosure of reviewer identity is with reference to reviewer comments included in decision letter(s). The following individuals involved in review of your submission have agreed to reveal their identity: Roshan Kumar (Reviewer #2)

Transaction Report:

DOI: <https://doi.org/10.1128/msystems.01019-21>

October 7, 2021

Prof. Bruno Gonzalez-Zorn
Universidad Complutense de Madrid
Avda Puerta de Hierro s/n
Madrid
Spain

Re: mSystems01019-21 (Dissemination routes of carbapenem and pan-aminoglycoside resistance mechanisms in hospital and urban wastewater canalizations of Ghana)

Dear Prof. Bruno Gonzalez-Zorn:

Thank you for submitting your manuscript to mSystems. We have completed our review and I am pleased to inform you that, in principle, we expect to accept it for publication in mSystems. However, acceptance will not be final until you have adequately addressed the reviewer comments.

Preparing Revision Guidelines

Sincerely,

Rup Lal

Editor, mSystems

Journals Department
Reviewer comments:

Reviewer #1 (Comments for the Author):

The manuscript entitled 'Dissemination routes of carbapenem and pan-aminoglycoside resistance mechanisms in hospital and urban wastewater canalizations of Ghana' by Delgado-Blas et al. is an extensive and well-planned study for deciphering the presence of drug-resistant microbes in hospital and urban wastewater canalizations of Ghana. The study surely has merit and the data generated is of great importance. Few changes need to be incorporated before the publication of this manuscript.

Comments:

1. Line 32: The authors must mention the number of resistant bacterial isolates they obtained after culturing.
2. Line 42: What is meant by genetic platforms?
3. Line 74: What is meant by 'consequence of the course of wastewater'? please reframe the sentence for better clarity.
4. The authors must provide the GPS locations of their sampling locations as supplementary information.
5. Line 135: The number of samples mentioned is 24. However, I find the figure confusing as per the details of the samples given in the manuscript. Please provide a proper breakup about the samples including the associated metadata if any in tabular form.
6. Line 142: 'remained higher' - Please provide a numerical/ percent value depicting the higher counts of resistant bacteria in wastewater after canalization before the hospital.
7. Line 184 -185 How many isolates of *Pseudomonas putida* were taken for the pan and core genome analysis?
8. Line 203: Please provide the number of *C. werkmanii* genomes taken for the pan-genome analysis.
9. Line 293: What is meant by 'independent structure'?
10. Line 296: Please provide the method for calculating the copy number of plasmids.
11. Line 471: change obtained 'in' to obtained 'from'
12. Line 478: The method for plating the samples for CFU counts seems improper. The manuscript mentions direct plating of 1ml sample onto MacConkey agar. The proper way is to make serial dilutions and then do CFU. Please provide a reference if direct plating was done.
13. A table summarising the genomic information of the bacterial isolates sequenced in the study with proper details regarding genome stats, sequencing methods, and the number of gene counts (related to antibiotic genes) should be incorporated.
14. Carefully check the manuscript for grammatical and typo errors.

Reviewer #2 (Comments for the Author):

The manuscript entitled " Dissemination routes of carbapenem and pan-aminoglycoside resistance mechanisms in hospital and urban wastewater canalizations of Ghana" by Delgado-Blas et al. analyzed the AMR, mobile genetic elements and bacterial populations in waste-water collected from hospital wastewater and urban waste treatment plant.

Overall the study is well planned and the methodology is robust. The result section should be more focused. Certainly, the Grammar can be improved.

Minor comments:

- Line 28: Mobile genetic platforms is not the right term.
- Line 31: Please mention the abbreviation after the urban waste treatment plant (UWTP).
- Line 79: Reframe the sentence.
- Line 90-93: More references should be added to emphasize this statement.
- Line 116: even to the
- Line 116-120: Reframe the sentence.
- Line 131: Duplicate samples from three different points doesn't sum up 24.
- Line 465: Instead "All wastewater sample", mention the number of samples.
- Line 471: "obtained in" should be "obtained at"
- Line 476: Reframe the sentence.
- Line 471-472: The calculation suggests that the number of samples was 18 and in line 477, the number of samples mentioned as 24. Make it clearer.
- Line 478-480: How did you count the CFU without doing any serial dilutions?
- Line 482; 482: How did you determine the concentration of antibiotics?
- Line 470: Please mention the MALDI-TOF cutoff value for bacterial identification.
- Line 496: Make it established.
- Line 502-508: Please provide a list of primers used in this section.
- Figure 1B: The data representation is not clear. Total bacteria does not include Carbapenem-resistant and Aminoglycoside-resistant?
- Figure 2: There is no symbolic representation of color code for the outer ring.

Response to Reviewer Comments

Please find enclosed our response to the referees' comments. We thank the reviewers for their help and for their comments, which have contributed to enhance this work.

Manuscript title: Dissemination routes of carbapenem and pan-aminoglycoside resistance mechanisms in hospital and urban wastewater canalizations of Ghana

Authors: Jose Delgado-Blas, Cecilia Valenzuela Agüi, Elena Marín Rodríguez, Carlos Serna, Natalia Montero, Courage Setsoafia Saba, and Bruno Gonzalez- Zorn

Reference: mSystems01019-21

Reviewer #1

The manuscript entitled 'Dissemination routes of carbapenem and pan-aminoglycoside resistance mechanisms in hospital and urban wastewater canalizations of Ghana' by Delgado-Blas *et al.* is an extensive and well-planned study for deciphering the presence of drug-resistant microbes in hospital and urban wastewater canalizations of Ghana. The study surely has merit and the data generated is of great importance. Few changes need to be incorporated before the publication of this manuscript.

Comments:

1. Line 32: The authors must mention the number of resistant bacterial isolates they obtained after culturing.

The number of carbapenem/pan-aminoglycoside resistant bacterial isolates selected for posterior analyses after culturing has been specified in the abstract:

Line 32: "From all carbapenem/pan-aminoglycoside resistant bacteria, 36 isolates were selected to determine bacterial species and phenotypical resistance profiles."

2. Line 42: What is meant by genetic platforms?

We refer to 'genetic platforms' as all genetic elements involved in the capture, mobilization and dissemination of resistance genes, including integrons, insertion sequences (ISs), transposons, plasmids and others. In order to clarify this point, the term 'genetic platforms' has been changed by 'mobile genetic elements' throughout the manuscript:

Line 43: "... were associated to MGEs (mobile genetic elements) that allowed for their dissemination between environmental and clinical bacterial hosts."

3. Line 74: What is meant by 'consequence of the course of wastewater'? please reframe the sentence for better clarity.

We meant the implications of wastewater flux as a vehicle for biological elements from the sources to other environments and populations through natural or artificial canalizations when wastewater is not treated. For clarity, the sentence has been reframed:

Line 76: “Furthermore, the resulting compounds and biological associations could reach the environment and, eventually, human and animal populations due to the flow of wastewater.”

4. The authors must provide the GPS locations of their sampling locations as supplementary information.

A supplementary table (Table S1) has been included with the manuscript, indicating the exact GPS coordinates of all sampling locations, as well as the type of wastewater canalization of each sampling point.

5. Line 135: The number of samples mentioned is 24. However, I find the figure confusing as per the details of the samples given in the manuscript. Please provide a proper breakup about the samples including the associated metadata if any in tabular form.

The paragraph has been rephrased for a better understanding on the number of samples:

Line 132: “In each of the four sampling locations, three hospitals (Tamale West Hospital-TWH; Tamale Central Hospital-TCH; and Tamale Teaching Hospital-TTH) and one Urban Waste Treatment Plant (UWTP) located in the metropolitan area of Tamale. Wastewater samples were obtained from three different points: the canalization before the hospital (the first treatment pond in the case of the UWTP), the drainage of the hospital onto the urban canalization (the last treatment pond of the UWTP) and the canalization after the hospital (the point where treated water is discharged to the environment after the UWTP) (Figure 1A, Table S1). From each point, duplicate samples were collected, for a total of 24 wastewater samples.”

In addition, following the reviewer’s suggestion, a breakup about the samples and the associated metadata is indicated in tabular form in supplementary Table S1.

Furthermore, a sentence has been also added to the ‘Sampling and bacterial isolation’ section of the manuscript Methods to clarify again the origin of the samples:

Line 553: “Therefore, a total of 24 wastewater samples were collected (4 sampling locations X 3 sampling points X 2 wastewater samples).”

6. Line 142: 'remained higher' - Please provide a numerical/ percent value depicting the higher counts of resistant bacteria in wastewater after canalization before the hospital.

The percent value of carbapenem/pan-aminoglycoside resistant bacteria with respect to the total bacteria in wastewater samples from canalizations before the hospital comparing with those from canalizations after the hospital has been included in the manuscript for each of the three sampled hospitals:

Line 145: “However, the counts of resistant bacteria in wastewater from canalizations after hospitals remained higher than those observed from canalizations before the hospital (Figure 1B): from 0.00069% to 0.013%, from 0.0% to 0.00036% and from 0.0262% to 0.133%”

carbapenem-/pan-aminoglycoside-resistant bacteria in the TWH, TCH and TTH, respectively.”

7. Line 184 -185 How many isolates of *Pseudomonas putida* were taken for the pan and core genome analysis?

The number of *P. putida* isolates included in the pan- and core-genome analyses has been included as suggested by the reviewer:

Line 217: “Pan-genome analysis of 11 *P. putida* isolates showed the genetic divergence between the two STs identified, ...”

8. Line 203: Please provide the number of *C. werkmanii* genomes taken for the pan-genome analysis.

The number *C. werkmanii* isolates included in the pan- and core-genome analyses has been included as suggested by the reviewer:

Line 252: “..., the 11 *C. werkmanii* isolates included in the pan-genome analysis showed a close relationship between all of them...”

9. Line 293: What is meant by 'independent structure'?

We refer to ‘independent structure’ as a circular genetic structure which is independent from the chromosome (extrachromosomal), similar to plasmid structures. The term ‘independent structure’ has been modified in the manuscript in order to clarify this concept:

Line 358: “Additionally, in two *C. werkmanii*, one *Escherichia coli* and one *Klebsiella pneumoniae*, the gene *bla*_{NDM-1} was located in a circular extrachromosomal genetic structure of 9.3kb.”

10. Line 296: Please provide the method for calculating the copy number of plasmids.

The plasmid copy number was estimated from the comparison of the mean coverage, mapping the short reads obtained from Illumina sequencing to specific genetic sequences, after quality control and trimming. In the case of the plasmoid structure, the copy number was firstly calculated by comparing the mean coverage of the *bla*_{NDM-1} gene (integrated in a single copy in the plasmoid) with the mean coverage of the sequence of the IncC origin of replication, the plasmid that was the most probable source of this genetic element. Then, the mean coverage of the whole 9.3kb plasmoid length was compared with respect to the mean coverage of the whole IncC plasmid length. Likewise, the copy numbers of both the IncC plasmid and the plasmoid structure with respect to the chromosome were obtained from the comparison of the mean coverage of the aforementioned genetic sequences with the mean coverage of the 16S rRNA gene and the mean coverage of the total chromosome length.

A clarifying sentence of the method for calculating the copy number has been added to the line indicated by the reviewer:

Line 361: “Furthermore, this ‘plasmoid’ structure presented an average copy number of ~3, comparing to the one copy of the IncC plasmid co-habiting in the same cell, according to the

comparison of their mean short-read coverage between them and with respect to the bacterial chromosome.”

In addition, a paragraph describing the method for calculating the copy number has been included in the ‘Analysis of 16S-RMTase and carbapenemase gene-carrying genomic structures’ section of the manuscript Methods to explain it in detail:

Line 699: “The copy number of particular 16S-RMTase/carbapenemase gene-carrying structures was estimated from the comparison of the mean coverage, calculated by mapping the QC filtered and trimmed short reads against the closed genome, of specific genetic signatures and the whole length of these structures and the bacterial chromosome.”

11. Line 471: change obtained 'in' to obtained 'from'

The word has been modified following the reviewer’s suggestion:

Line 546: “In each of the four sampling locations, three duplicate samples of 50ml were obtained from three different points...”

12. Line 478: The method for plating the samples for CFU counts seems improper. The manuscript mentions direct plating of 1ml sample onto MacConkey agar. The proper way is to make serial dilutions and then do CFU. Please provide a reference if direct plating was done.

The reviewer is right. All wastewater samples were processed by 10-fold serial dilutions before plating them. The samples were serially diluted 10-fold in PBS and 100µl from each dilution was plated onto the MacConkey agar plates without antibiotic pressure and MacConkey agar plates supplemented with 8mg/l imipenem and with 200 mg/l gentamicin plus 200 mg/l amikacin. Then, bacterial counts were carried out from these plates in order to obtain the CFU/ml for both total bacteria and carbapenem-/pan-aminoglycoside-resistant bacteria. The method for plating the samples and performing CFU counts has been corrected and extended in the ‘Sampling and bacterial isolation’ section of the manuscript Methods:

Line 555: “To determine the general bacterial population from each sampling point, wastewater samples were serially diluted 10-fold in PBS and 100µl from each dilution was plated onto MacConkey agar plates with no antibiotic pressure (Oxoid Ltd., Basingstoke, Hampshire, UK), calculating the number of CFU/ml per sample. For counting and isolation of pan-aminoglycoside-resistant bacteria, 100µl from each of the aforementioned dilutions were plated onto MacConkey agar plates supplemented with 200 mg/l gentamicin and 200 mg/l amikacin (Sigma-Aldrich Inc., Saint Louis, Missouri, USA). Likewise, carbapenem-resistant bacteria were counted and isolated by plating 100 µl from each 10-fold dilution onto MacConkey agar plates supplemented with 8mg/l imipenem (Sigma-Aldrich Inc.).”

13. A table summarising the genomic information of the bacterial isolates sequenced in the study with proper details regarding genome stats, sequencing methods, and the number of gene counts (related to antibiotic genes) should be incorporated.

The tables required by the reviewer were already included in the manuscript as supplementary material, not only summarizing the bioinformatic tools and stats regarding the

genome assemblies of bacterial isolates sequenced, but also the raw data obtained from the different WGS platforms:

Table S2. Metadata and data of raw reads from Nanopore long-read whole genome sequencing of carbapenem and pan-aminoglycoside resistant bacteria from wastewater canalizations of Tamale metropolitan area.

Table S3. Metadata and data of raw reads from Illumina short-read whole genome sequencing of carbapenem and pan-aminoglycoside resistant bacteria from wastewater canalizations of Tamale metropolitan area.

Table S4. Metadata and data of SPAdes assemblies from short-read whole genome sequencing of carbapenem and pan-aminoglycoside resistant bacteria from wastewater canalizations of Tamale metropolitan area.

Table S5. Metadata and data of Unicycler hybrid assemblies from short- and long-read whole genome sequencing of carbapenem and pan-aminoglycoside resistant bacteria from wastewater canalizations of Tamale metropolitan area.

In addition, as it was indicated in all this supplementary material: “All sequencing data are encompassed under the umbrella project PRJEB39000 in the European Nucleotide Archive (ENA).”

Regarding the number of genes related to antibiotic resistance, it is shown in the ‘Figure 2. Carbapenem-/pan-aminoglycoside-resistant bacteria data’ of the manuscript, together with the specific resistance genes to all antibiotic classes that were found in all bacterial isolates sequenced.

14. Carefully check the manuscript for grammatical and typo errors.

The manuscript has been thoroughly revised and the grammatical and typo errors have been carefully checked and corrected by English-speaking collaborators.

Reviewer #2

The manuscript entitled " Dissemination routes of carbapenem and pan-aminoglycoside resistance mechanisms in hospital and urban wastewater canalizations of Ghana" by Delgado-Blas et al. analyzed the AMR, mobile genetic elements and bacterial populations in waste-water collected from hospital wastewater and urban waste treatment plant. Overall the study is well planned and the methodology is robust. The result section should be more focused. Certainly, the Grammar can be improved.

Minor comments:

Line 28: Mobile genetic platforms is not the right term.

We refer to ‘genetic platforms’ as all genetic elements involved in the capture, mobilization and dissemination of resistance genes, including integrons, insertion sequences (ISs), transposons, plasmids and others. However, following the reviewers’ suggestion, the term ‘genetic platforms’ has been changed by ‘mobile genetic elements’ throughout the manuscript:

Line 28: “Therefore, the present study analyzed the epidemiological scenario of resistance genes, mobile genetic elements (MGEs) and bacterial populations in wastewater around the Tamale metropolitan area (Ghana).”

Line 31: Please mention the abbreviation after the urban waste treatment plant (UWTP).

The abbreviation UWTP after ‘urban waste treatment plant’ has been included:

Line 31: “Wastewater samples were collected from the drainage and the canalizations before and after three hospitals and one urban waste treatment plant (UWTP).”

Line 79: Reframe the sentence.

The sentence has been reframed:

Line 78: “However, low- and middle-income countries (LMICs) largely lack the infrastructures and systems to carry out the accurate processing of wastewater, not only from the community, but also from hospitals and health-care settings, which entails a greater health risk.”

Line 90-93: More references should be added to emphasize this statement.

More references have been included in the manuscript in order to support the statement about the efficacy of carbapenems and aminoglycosides against MDR *Enterobacteriaceae* infections with limited therapeutic options (line 94):

9. Tamma PD, Cosgrove SE, Maragakis LL. 2012. Combination therapy for treatment of infections with gram-negative bacteria. *Clin Microbiol Rev* 25(3):450–470. doi:10.1128/CMR.05041-11.

10. Daikos GL, Tsaousi S, Tzouveleki LS, Anyfantis I, Psychogiou M, Argyropoulou A, Stefanou I, Sypsa V, Miriagou V, Nepka M, Georgiadou S, Markogiannakis A, Goukos D, Skoutelis A. 2014. Carbapenemase-producing *Klebsiella pneumoniae* bloodstream infections: lowering mortality by antibiotic combination schemes and the role of carbapenems. *Antimicrob Agents Chemother* 58(4):2322–2328. doi:10.1128/AAC.02166-13.

Line 116: even to the

It has been modified in the manuscript:

Line 116: “...even to the last-resort aminoglycoside, plazomicin”.

Line 116-120: Reframe the sentence.

The sentence has been reframed in order to clarify these concepts:

Line 117: “The association of some 16S-RMTase genes with diverse MGEs has increased their potential for spread over the last decade, even between bacteria from human, animal and environmental sources. Furthermore, these MGEs frequently co-integrate 16S-RMTase genes with other resistance determinants, especially β -lactamase genes, due to the broad use of aminoglycoside- β -lactam combinations in human medicine.”

Line 131: Duplicate samples from three different points doesn't sum up 24.

The paragraph has already been rephrased for a better understanding on the number of samples, also in accordance with the suggestion of reviewer #1:

Line 132: “In each of the four sampling locations, three hospitals (Tamale West Hospital-TWH; Tamale Central Hospital-TCH; and Tamale Teaching Hospital-TTH) and one Urban Waste Treatment Plant (UWTP) located in the metropolitan area of Tamale. Wastewater samples were obtained from three different points: the canalization before the hospital (the first treatment pond in the case of the UWTP), the drainage of the hospital onto the urban canalization (the last treatment pond of the UWTP) and the canalization after the hospital (the point where treated water is discharged to the environment after the UWTP) (Figure 1A, Table S1). From each point, duplicate samples were collected, for a total of 24 wastewater samples.”

Line 465: Instead "All wastewater sample", mention the number of samples.

It has been modified in the manuscript:

Line 540: “Twenty-four wastewater samples from the different locations were collected in August 2017 in the metropolitan area of Tamale (Ghana).”

Line 471: "obtained in" should be "obtained at"

The word has been already modified following the suggestion of reviewer #1:

Line 547: “From each location, three duplicate samples of 50ml were obtained from three different points...”).”

Line 476: Reframe the sentence.

The sentence has been reframed in order to simplify it:

Line 551: “The three sampling points within the same sampling location were at least 100m apart from each other to ensure a sufficient distance between different wastewater canalizations.”

Line 471-472: The calculation suggests that the number of samples was 18 and in line 477, the number of samples mentioned as 24. Make it clearer.

The paragraph has been modified and extended to explain in detail the total number of samples and their origin:

Line 546: “In each of the four sampling locations, three duplicate samples of 50ml were obtained from three different points: the wastewater urban canalization before the hospital (the first treatment pond in the case of the UWTP), the drainage of the hospital onto the urban canalization (the last treatment pond of the UWTP) and the wastewater urban canalization after the hospital (the point where treated water is discharged to the environment after the UWTP).”

Line 553: “Therefore, a total of 24 wastewater samples were collected (4 sampling locations X 3 sampling points X 2 wastewater samples).”

Line 478-480: How did you count the CFU without doing any serial dilutions?

All wastewater samples were processed by 10-fold serial dilutions before plating them. The method for plating the samples and performing CFU counts has been corrected and extended in the ‘Sampling and bacterial isolation’ section of the manuscript Methods:

Line 555: “To determine the general bacterial population from each sampling point, wastewater samples were serially diluted 10-fold in PBS and 100µl from each dilution was plated onto MacConkey agar plates with no antibiotic pressure (Oxoid Ltd., Basingstoke, Hampshire, UK), calculating the number of CFU/ml per sample. For counting and isolation of pan-aminoglycoside-resistant bacteria, 100µl from each of the aforementioned dilutions were plated onto MacConkey agar plates supplemented with 200 mg/l gentamicin and 200 mg/l amikacin (Sigma-Aldrich Inc., Saint Louis, Missouri, USA). Likewise, carbapenem-resistant bacteria were counted and isolated by plating 100 µl from each 10-fold dilution onto MacConkey agar plates supplemented with 8mg/l imipenem (Sigma-Aldrich Inc.)”

Line 482: How did you determine the concentration of antibiotics?

In the case of aminoglycosides, both the specific aminoglycoside compounds (gentamicin and amikacin) and their concentrations were established in accordance with the aminoglycoside pressure required to select pan-aminoglycoside resistant bacteria, meaning bacteria producing 16S rRNA methyltransferases. The use of other aminoglycoside combinations or lower concentrations of these antibiotics can potentially lead to the selection of bacteria expressing other resistance mechanisms, especially aminoglycoside-modifying enzymes. This selection method has been extensively applied and it was explained in the work:

Hidalgo L, Hopkins KL, Gutierrez B, Ovejero CM, Shukla S, Douthwaite S, Prasad KN, Woodford N, Gonzalez-Zorn B. 2013. Association of the novel aminoglycoside resistance determinant RmtF with NDM carbapenemase in *Enterobacteriaceae* isolated in India and the UK. *J Antimicrob Chemother* 68(7):1543–1550. doi:10.1093/jac/dkt078.

Regarding the carbapenems, imipenem has a broader spectrum comparing with other carbapenems and it is effective against *Enterobacteriaceae*, *Pseudomonas* spp. and *Acinetobacter baumannii*, among others, being relevant for therapy of septicemia, post-operative sepsis or nosocomial pneumonia causing by these bacteria. The MIC breakpoint to consider one *Enterobacteriaceae* or *Pseudomonas* species clinically resistant to imipenem is >4mg/L. However, combinations of an ESBL or AmpC enzyme and impermeability confer reduced susceptibility to imipenem in *Enterobacteriaceae*, and porin loss and alteration in efflux pumps may also reduce imipenem susceptibility in *Pseudomonas* spp. Thus, in order to exclusively select carbapenemase-producing bacteria, which are able to resist high concentrations of carbapenems, the imipenem concentration applied was double the MIC breakpoint value (8mg/L). This selection criterium was established following previous works and the guidelines of the European Committee on Antimicrobial Susceptibility Testing (EUCAST):

European Committee on Antimicrobial Susceptibility Testing. 2019. Breakpoint tables for interpretation of MICs and zone diameters. Version 9.0.

https://www.eucast.org/fileadmin/src/media/PDFs/EUCAST_files/Breakpoint_tables/v_9.0_Breakpoint_Tables.pdf

https://www.eucast.org/resistance_mechanisms/

European Committee on Antimicrobial Susceptibility Testing. 2017. EUCAST guideline for the detection of resistance mechanisms and specific resistances of clinical and/or epidemiological importance. Version 2.0.

https://www.eucast.org/fileadmin/src/media/PDFs/EUCAST_files/Resistance_mechanisms/EUCAST_detection_of_resistance_mechanisms_170711.pdf

The references of the works describing the selection methods for carbapenem and pan-aminoglycoside resistant bacteria have been included in the ‘Sampling and bacterial isolation’ section of the manuscript Methods (lines 561 and 563, respectively).

Line 490: Please mention the MALDI-TOF cutoff value for bacterial identification.

The MALDI-TOF cutoff value for bacterial identification has been included in the manuscript:

Line 568: “Bacterial species determinations were carried out by MALDI-TOF mass spectrometry in Centro de Vigilancia Sanitaria Veterinaria (VISAVET Health Surveillance Centre, Madrid, Spain), with a cut-off value of ≥ 2.3 for accurate species identification.”

Line 496: Make it established.

The word has been modified in the manuscript:

Line 576: “...for antibiotics and bacterial species with no established breakpoints in EUCAST documents...”

Line 502-508: Please provide a list of primers used in this section.

The works referenced in lines 582-588 provide all necessary information to replicate the 16S-RMTase and carbapenemase gene screenings by PCR, including primer sequences, melting temperatures, product sizes for all inspected genes and the original studies from which the PCR protocols were obtained.

References:

39. Bado I, Papa-Ezdra R, Delgado-Blas JF, Gaudio M, Gutiérrez C, Cordeiro NF, García-Fulgueiras V, Araújo Pirez L, Seija V, Medina JC, Rieppi G, Gonzalez-Zorn B, Vignoli R. 2018. Molecular Characterization of Carbapenem-Resistant *Acinetobacter baumannii* in the Intensive Care Unit of Uruguay’s University Hospital Identifies the First *rmtC* Gene in the Species. *Microb Drug Resist* 24:1012–1019.

40. Dallenne C, Da Costa A, Decré D, Favier C, Arlet G. 2010. Development of a set of multiplex PCR assays for the detection of genes encoding important beta-lactamases in *Enterobacteriaceae*. *J Antimicrob Chemother* 65:490–495.

41. Murali S, Jambulingam M, Tiru V, Kulanthai LT, Rajagopal R, Padmanaban P, Madhavan HN. 2012. A study on isolation rate and prevalence of drug resistance among microorganisms isolated from multiorgan donor and donor corneal rim along with a report on existence of *bla*_{NDM-1} among Indian population. *Curr Eye Res* 37:195–203.

42. Batchelor M, Hopkins K, Threlfall EJ, Clifton-Hadley FA, Stallwood AD, Davies RH, Liebana E. 2005. *bla*_{CTX-M} genes in clinical *Salmonella* isolates recovered from humans in England and Wales from 1992 to 2003. *Antimicrob Agents Chemother* 49:1319–1322.

Figure 1B: The data representation is not clear. Total bacteria does not include Carbapenem-resistant and Aminoglycoside-resistant?

The reviewer is right. Bar charts of Figure 1B show the bacterial counts (CFU/ml) in logarithmic scale calculated from the same wastewater dilutions plated in MacConkey agar plates with and without antibiotics. Therefore, total bacteria counts estimated from the MacConkey agar plates without antibiotics also included those bacteria from the same sample that were resistant to some or all the antibiotics with which the plates for resistant bacteria counts were supplemented (gentamicin and amikacin and/or imipenem). In order to clarify this point, a sentence has been added to the legend of the Figure 1B:

Line 166: “**1B:** Bacterial counts (CFU/ml) of total bacteria, carbapenem-resistant bacteria and pan-aminoglycoside-resistant bacteria in wastewater samples from canalizations of Tamale. Total bacteria encompass the entire bacterial population, including carbapenem and pan-aminoglycoside resistant bacteria.”

Figure 2: There is no symbolic representation of color code for the outer ring.

The outer ring colors show different clusters in accordance with the bacterial species. This was previously specified in the legend of the figure:

Line 197: “**Figure 2.** Tree of carbapenem-/pan-aminoglycoside-resistant isolates based on its total resistance gene content. Bacterial species are indicated by leaf labels and outer ring colored sections.”

Following the reviewer’s suggestion, a symbolic representation for the outer ring has been included in the figure:

Yours sincerely,

BRUNO GONZALEZ ZORN

Bruno González Zorn

Head of the Antimicrobial Resistance Unit (ARU)
Faculty of Veterinary Medicine and VISAVET Health Surveillance Centre
Complutense University of Madrid

January 2, 2022

Prof. Bruno Gonzalez-Zorn
Universidad Complutense de Madrid
Avda Puerta de Hierro s/n
Madrid
Spain

Re: mSystems01019-21R1 (Dissemination routes of carbapenem and pan-aminoglycoside resistance mechanisms in hospital and urban wastewater canalizations of Ghana)

Dear Prof. Bruno Gonzalez-Zorn:

Your manuscript has been accepted, and I am forwarding it to the ASM Journals Department for publication. For your reference, ASM Journals' address is given below. Before it can be scheduled for publication, your manuscript will be checked by the mSystems senior production editor, Ellie Ghatineh, to make sure that all elements meet the technical requirements for publication. She will contact you if anything needs to be revised before copyediting and production can begin. Otherwise, you will be notified when your proofs are ready to be viewed.

Publication Fees:

We recognize that the video files can become quite large, and so to avoid quality loss ASM suggests sending the video file via <https://www.wetransfer.com/>. When you have a final version of the video and the still ready to share, please send it to mssystemsjournal@msubmit.net.

Sincerely,

Rup Lal
Editor, mSystems

Journals Department
Table S5: Accept
Table S1: Accept
Figure S1: Accept
Table S2: Accept
Figure S3: Accept
Figure S2: Accept
Table S6: Accept
Table S3: Accept
Figure S4: Accept
Table S4: Accept